# Prevalence and mechanisms of somatic deletions in single human neurons during normal aging and in DNA repair disorders

Junho Kim[1,2,3,4,5], August Yue Huang [1,2,3,4], Shelby L. Johnson[6], Jenny Lai [1,2,3,4], Laura Isacco[1,2,3,4,7,8], Ailsa M. Jeffries[6], Michael B. Miller [1,2,3,4,7,8,9], Michael A. Lodato[1,2,3,4,6,7,8], Christopher A. Walsh [1,2,3,4,7,8] ✉ & Eunjung Alice Lee [1,2,3,4] ✉

Replication errors and various genotoxins cause DNA double-strand breaks (DSBs) where error-prone repair creates genomic mutations, most frequently focal deletions, and defective repair may lead to neurodegeneration. Despite its pathophysiological importance, the extent to which faulty DSB repair alters the genome, and the mechanisms by which mutations arise, have not been systematically examined reflecting ineffective methods. Here, we develop PhaseDel, a computational method to detect focal deletions and characterize underlying mechanisms in single-cell whole genome sequences (scWGS). We analyzed high-coverage scWGS of 107 single neurons from 18 neurotypical individuals of various ages, and found that somatic deletions increased with age and in highly expressed genes in human brain. Our analysis of 50 single neurons from DNA repair-deficient diseases with progressive neurodegeneration (Cockayne syndrome, Xeroderma pigmentosum, and Ataxia telangiectasia) reveals elevated somatic deletions compared to age-matched controls. Distinctive mechanistic signatures and transcriptional associations suggest roles for somatic deletions in neurodegeneration.

DNA double-strand breaks (DSBs) are among the most damaging DNA lesions, rapidly triggering DNA repair or apoptosis[1,2]. Dividing human fibroblasts are known to accumulate ~50 DSBs per cell cycle[3], and common environmental sources such as ionizing radiation or ultraviolet light continuously produce DSBs in human cells[2]. DSBs have been implicated in not only playing an essential physiological role in human neurons[4], but also causing neurodegeneration when misrepaired[5,6]. Despite the significant pathophysiological implications of DSB and its repair, the extent to which DSBs occur and the mechanisms by which they are repaired have not been systematically examined in

human neurons of aging and DSB repair-defective individuals mainly due to a lack of effective means[4–6].

To maintain homeostasis against hazardous DSBs, distinct DSB repair mechanisms have evolved in different cellular contexts[7]. Many DSB repair mechanisms require end processing of two DSBs to ligate them, resulting in the modification or loss of the genomic sequences around the breakpoints[2,7]. These genomic changes create DSB-induced structural variants (SVs) and may influence cellular functions. Homologous recombination, a major error-free DSB repair activated during cell replication cannot be utilized in fully differentiated and thus non-

[1]Division of Genetics and Genomics, Boston Children's Hospital, Boston, MA, USA. [2]Manton Center for Orphan Disease, Boston Children's Hospital, Boston, MA, USA. [3]Department of Pediatrics, Harvard Medical School, Boston, MA, USA. [4]Broad Institute of MIT and Harvard, Cambridge, MA, USA. [5]Department of Biological Sciences, Sungkyunkwan University, Suwon, South Korea. [6]Department of Molecular, Cell and Cancer Biology, University of Massachusetts Medical School, Worcester, MA, USA. [7]Howard Hughes Medical Institute, Boston Children's Hospital, Boston, MA, USA. [8]Department of Neurology, Harvard Medical School, Boston, MA, USA. [9]Department of Pathology, Brigham and Women's Hospital, Harvard Medical School, Boston, MA, USA. ✉e-mail: christopher.walsh@childrens.harvard.edu; EAlice.Lee@childrens.harvard.edu

replicating post-mitotic neurons[1]. Thus, other error-prone DSB repair mechanisms are likely to induce more SVs in post-mitotic neurons than other cell types. Careful examination of SV breakpoints and their sequence features is essential to understand the impact of SVs and underlying mutational and DSB repair processes. However, most related efforts have been made only in animal models with artificial DSB induction[8,9], and few studies have been done with human neurons.

Advances in single-cell genomics now provide a way to detect various types of somatic mutations in human post-mitotic neurons. Recent single-cell whole-genome sequencing (scWGS) studies have reported the occurrence of hundreds to thousands of somatic single-nucleotide variants (sSNVs)[10–12], tens of megabase-scale somatic copy number variants (CNVs)[13–17], and less than one somatic L1 insertion in each neuron of neurotypical brains[18–21]. However, the analysis of single-neuronal SVs, which are closely related to mis-repaired DSBs, has still remained a challenge, due to the substantial SV-like chimeric artifacts that hinder the detection of true SVs arising during whole-genome amplification (WGA) of a single cell[20,22]. Thus far, studies on somatic SVs in single neurons have only identified megabase-scale large deletions and duplications[13–17]. These previous studies created shallow-depth (<1×) scWGS data with read-depth-based analysis using large genomic bins, which is not sensitive enough to detect focal events and to determine exact breakpoints at single-base resolution[13,14,16]. For high-coverage scWGS, there has been no adequate bioinformatic method to distinguish true somatic SVs from prevalent artifacts.

In this study, we develop a computational method to detect focal somatic deletions, the most common form of SV emerging from mis-repaired DSBs, and characterize underlying DSB repair mechanisms by analyzing deletion breakpoints at single-base resolution. We analyzed 107 single neurons from 18 neurotypical individuals of various ages to assess the deletion accumulation with age and the contributions of underlying repair mechanisms. An increase in somatic deletions was identified with age and gene expression level, the latter occurring only with a specific repair mechanism. To test the hypothesis that accumulated DSBs are potentially implicated in contributing to neuronal degenerations, we also analyzed 50 single neurons from patients with three different neurodegenerative diseases with defective DNA repair —ataxia telangiectasia (AT), Cockayne syndrome (CS), and xeroderma pigmentosum (XP). AT is associated with defective DSB repair while CS and XP are associated with defective nucleotide-excision repair (NER). Our analysis reveals significant increases in somatic deletions not only for AT but also for the two NER diseases, suggesting a potential common underlying mechanism contributing to neurodegeneration.

## Results

### PhaseDel detects somatic nanodeletions in single human neurons

In scWGS data, uneven amplification and chimera formation during WGA results in SV-like artifacts that render conventional SV detection methods inapplicable as they are unable to discriminate genuine deletions from chimeric artifacts[20,22]. To accomplish the discrimination of the two with high confidence, we developed a computational method called PhaseDel. PhaseDel utilizes read-based linkage information between deletion breakpoints and nearby germline heterozygous single-nucleotide polymorphisms (SNPs) to delineate true deletions (both somatic and germline) based on the phasing patterns of spanning reads (Fig. 1a), a similar strategy adopted from our previous sSNV analysis[10,12,23]. PhaseDel utilizes two different and complementary callers, GATK[24] and DELLY2[25], to identify initial candidates that are validated by phasing analysis, allowing us to discover a wide range of deletions (Fig. 1b and Supplementary Fig. S1a).

We applied PhaseDel to a high-depth scWGS dataset (~45× on average) from a total of 107 prefrontal cortex (PFC) neurons of 18 neurotypical control individuals obtained from previously published studies[10,12], in which the genomic DNA was amplified using multiple displacement amplification (MDA). Figure 1c demonstrates one example of our phased somatic deletion candidates showing depletion of read depth and deletion-supporting reads perfectly linked with the heterozygous SNP, which were observed only in the scWGS data and not in the bulk WGS data from the cerebellum of the same individual. We identified a total of 1751 somatic deletions (3 bp–100 Kbp, mean 155 bp, median 4 bp) (Fig. 1d and Supplementary Fig. S1b, c). Most somatic (96.2%) and germline (99.5%) deletions were smaller than 100 bp, and somatic deletions were overall larger than germline deletions ($P = 2.2 \times 10^{-16}$, two-sided Mann–Whitney $U$ test; Fig. 1d and Supplementary Fig. S1d). We call the deletions targeted in this study nanodeletions as they were more than 1000 times smaller than microdeletion of megabase ranges[26].

We evaluated the sensitivity of our phasing method using phaseable pairs of germline heterozygous SNPs and germline heterozygous deletions. Unlike sSNV analysis, a high false negative rate (~30% on average, Supplementary Fig. S1e) was obtained even for known annotated deletions due to a small number of unphased reads caused by mapping errors or mis-clipping. Therefore, we allowed one unphased read to reduce false negatives, which despite increasing the false discovery rate (FDR) achieved 85.5% sensitivity per cell, on average (Fig. 1e).

Since phasing analysis can only be applied to ~25% of all deletion candidates that are close enough to germline SNPs (Supplementary Fig. S1f), extrapolation is required to estimate the genome-wide somatic deletion rate. We developed an estimation process based on a two-component model: true mutations and errors. This enabled not only extrapolating the genome-wide rate but also controlling the FDR at a similar level (<10%) by estimating the error rate per cell (Supplementary Fig. S1g). On average, we obtained 174.7 (±102.9 SD) somatic deletions per cell genome-wide from 107 single neurons from control individuals, estimated from an average of 16.4 (±13.6 SD) phased deletions per cell. An overview of the PhaseDel workflow is shown in Supplementary Fig. S2a.

### Evaluation of PhaseDel accuracy and validation using ultra-deep amplicon sequencing

It is difficult to assess the accuracy of PhaseDel using regular scWGS data since most somatic deletions are single-cell-specific, and it is impossible to obtain independent, not amplified DNA from the same single cell for validation. To overcome this challenge, we tested PhaseDel on published genome sequencing data of single-fibroblast-derived clones[27], consisting of two MDA-amplified-scWGS datasets from single fibroblasts (IL-11 and -12) and unamplified bulk WGS data of their kindred clone (IL-1c) (Supplementary Fig. S3a). Since all three WGS datasets are expected to share somatic mutations originating from the seed single fibroblast (Supplementary Fig. S3a, red star), we assessed the accuracy of PhaseDel by checking if each phased deletion candidate from the scWGS data was observed in the bulk WGS data of the kindred clone (see Methods). A total of 43/46 (93.5%) and 18/19 (94.7%) scWGS-derived PhaseDel deletions were confirmed in the bulk clone data (Supplementary Fig. S3b). We also evaluated if the genome-wide deletion count estimated by PhaseDel from the scWGS is comparable to the actual deletion count from the bulk clone data. Through multiple steps of conservative filtering (Supplementary Fig. S3c, see Methods), we obtained 452 somatic deletion candidates from the kindred bulk WGS data, which was comparable to the PhaseDel-estimated rates of 468.66 and 407.25 somatic deletions per fibroblast from the scWGS data (Supplementary Fig. S3d).

Next, we experimentally validated somatic deletion candidates from our single-neuron WGS data by designing custom amplicons that covered deletion breakpoints to directly capture the sequences spanning breakpoints. Out of 2585 high-confidence somatic deletion candidates from 107 normal and 50 disease neurons, we randomly selected a total of 244 deletions across all individuals and mechanisms

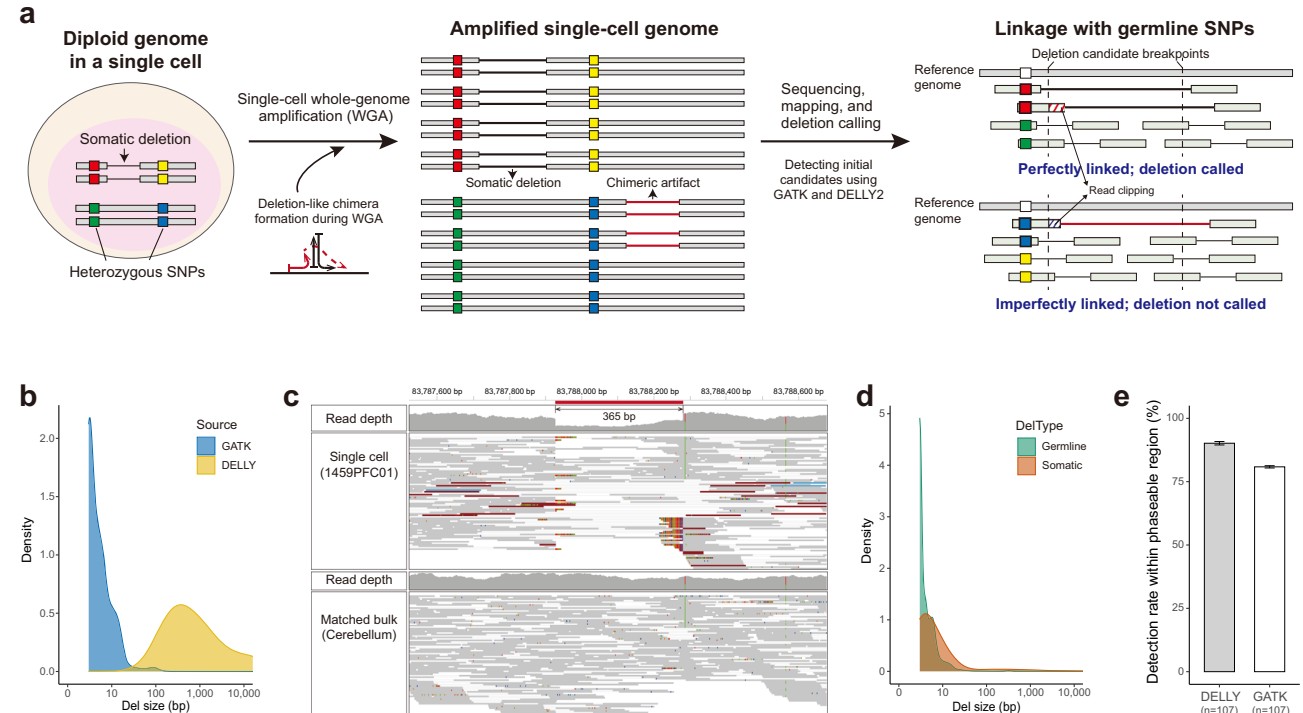

**Fig. 1 | Linkage-based detection of somatic deletions in single-cell WGS data.** **a** Schematic of PhaseDel. A true somatic deletion (black line) shows complete linkage with one allele (red) of a nearby heterozygous SNP whereas a deletion-like amplification artifact (red line) shows incomplete linkage with a nearby SNP allele (blue). PhaseDel utilizes raw calls from GATK and DELLY2 methods and makes final calls based on the linkage patterns of the initial deletion candidates. Box filled with diagonal lines represents clipped part of the read. **b** Deletion size distribution of two initial callers, GATK and DELLY2. **c** Integrative Genomics Viewer (IGV) screenshots of an example somatic deletion of 365 bp identified by PhaseDel. Upper and lower tracks demonstrate mapped reads from single-cell and bulk WGS data from the same individual, respectively. A reference allele of a heterozygous SNP on the right side of the deletion (red bar in the read depth track and no display in the WGS track) shows perfect linkage with the deletion-supporting reads (e.g., clipped reads). The deleted genomic region clearly shows a read depth decrease. **d** The size of somatic and germline deletions detected from 107 single neurons of 18 normal individuals. **e** The fraction of gold-standard germline deletions in phaseable regions detected by PhaseDel. *n*, number of single neurons; bar graph, mean ± 95% confidence interval (CI). Source data are provided as a Source Data file.

(Fig. 2a, see Methods). Genomes from four different sources (MDA-amplified DNA from the called cell and from an uncalled negative control cell, bulk PFC DNA and bulk non-brain DNA from the same individual) were amplified and indexed, then pooled and sequenced together with an average depth over 50,000×. With this amplicon sequencing, we validated 209/244 (85.7%) identified breakpoints from the MDA-amplified DNA of the single cells (Supplementary Fig. S4a, b). We failed to validate 35 candidates, including five with deletion-supporting reads in the negative control. For the remaining 30 failed candidates, we did not obtain any breakpoint-supporting reads from MDA-amplified DNA, and they did not show any differences in sequencing features compared to the validated ones, such as deletion types, supporting read count, genomic context, and homology sequence length. They are likely due to the discrepant sampling of deletion-supporting reads between the scWGS and amplicon sequencing data. Validation rates varied among individuals rather than disease status (Supplementary Fig. S4a, b).

Conservatively, this validation approach might not be able to distinguish single-cell-specific somatic deletions from chimeric artifacts that formed very early during MDA, but the validity of true somatic deletions could be further supported if the same events are clonally observed from amplicon sequencing reads of unamplified bulk DNA from the same donor. We found two ultra-low-level mosaic deletions that were supported by the bulk reads, which showed the exact same breakpoints as in the single cells (Fig. 2b, c). The first deletion was a short 45 bp deletion with supporting reads from the bulk PFC (0.3% variant allele frequency; VAF) and the bulk cerebellum DNA (0.5% VAF), initially detected by GATK (Fig. 2b). The other deletion was a DELLY-derived 160 bp deletion with supporting reads from

the bulk PFC (0.1% VAF) and absent in the bulk kidney DNA. Unlike the first deletion, this second deletion reflected a 7-bp microhomology between the two breakpoints, representing different underlying DSB repair mechanisms between the two mosaic deletions (Fig. 2c).

## Somatic nanodeletions increase with age and reflect distinctive underlying repair mechanisms

Genomic context, such as the length of sequence homology, at the deletion breakpoints provides fundamental information about the underlying mechanisms generating the deletion[28–31]. In PhaseDel, we implemented a module to predict which of six different mutational mechanisms was most likely to generate a candidate nanodeletion (see Methods), following criteria in a previous cancer genome study[31]. We applied the module to the somatic and germline deletion calls from 107 single neurons of normal individuals. In germline deletions, we observed contributions from underlying mechanisms similar to those identified in the previous work for the same size deletions (>100 bp), including major contributions from transposable element insertions (TEIs)[31,32]. This confirmed the validity of our developed module (Fig. 3a, upper left).

By contrast, somatic deletions in the same size range (>100 bp) in single human neurons revealed the contribution of mechanisms distinct from somatic deletions in cancer genomes (Fig. 3a, upper right). The major difference was the absence of contribution of fork stalling and template switching (FoSTeS), which accounted for ~20% of somatic deletions in the previous cancer study[31]. During DNA replication, the replication at the fork can stall and DNA polymerase can switch the template via microhomology of nearby single-stranded DNA, creating complex SVs[33]. The lack of deletions from the FoSTeS

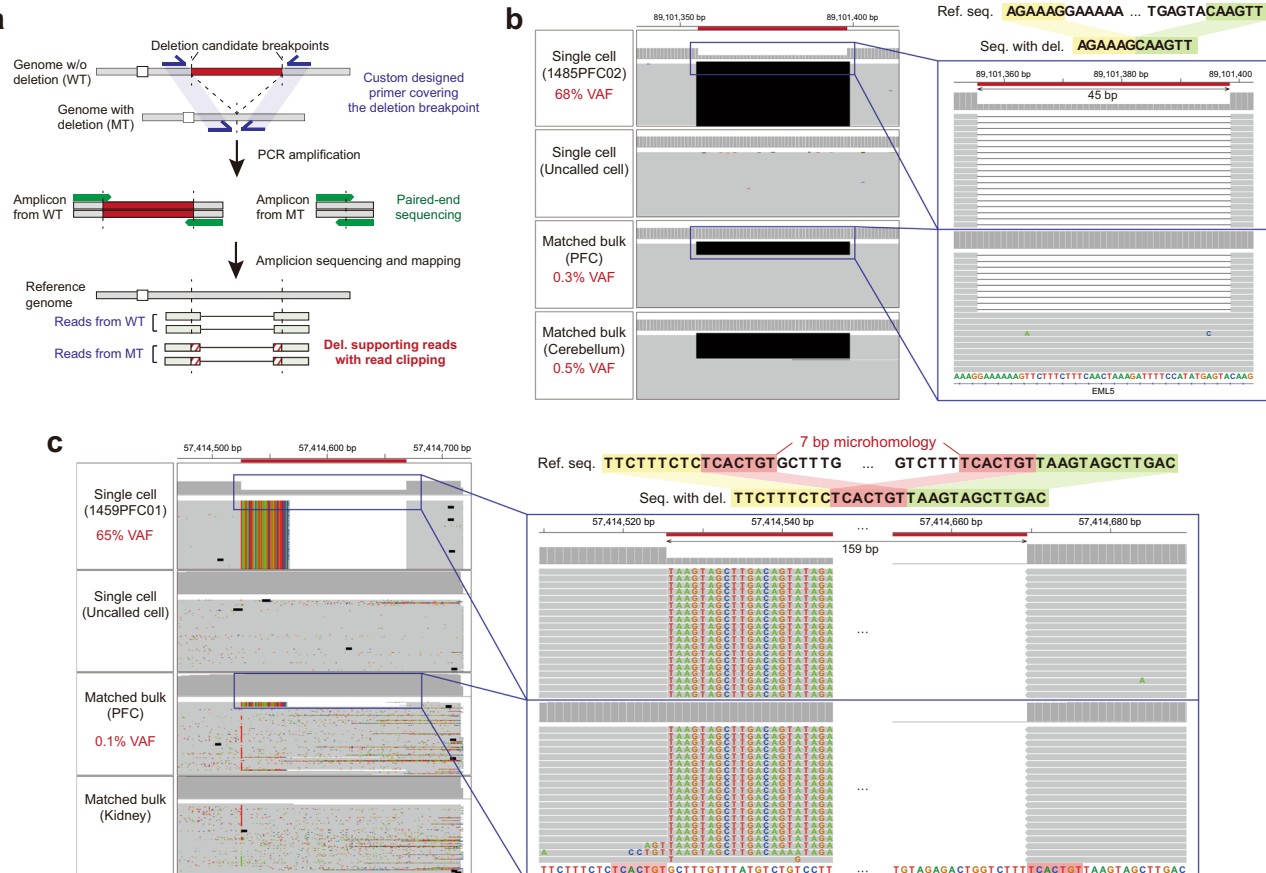

**Fig. 2 | Validation of somatic deletions by ultra-deep amplicon sequencing.**
**a** Illustration of validation sequencing with custom amplicons targeting predicted deletion breakpoints. Box filled with red diagonal lines represents clipped part of the read. WT wild-type, MT mutant. **b** Example of a validated low-level mosaic deletion. An IGV screenshot shows reads supporting the breakpoints of a 45-bp deletion (black line) in amplicon sequencing datasets of MDA-amplified DNA from the same single-cell from which PhaseDel called the deletion (VAF 68%) and bulk DNA from PFC and cerebellum of the same donor (VAF 0.3% and 0.5%) whereas MDA-amplicon sequencing of a different cell did not show any supporting read (second track). **c** Another validated mosaic deletion of 159 bp with 7 bp microhomology. Flanking reference genome sequences around deletion breakpoints are shown together (shaded by yellow and green for the left and right side of the breakpoint). Shared subsequence on both sides of the deletion breakpoint (microhomology) is shaded in red.

mechanism can be explained by the fact that our call sets were from non-replicating single mature neurons. In the set of all somatic deletions, non-homologous end-joining (NHEJ) and microhomology-mediated end-joining (MMEJ, also known as alternative end-joining) were the dominant mechanisms, accounting for 57.6% and 21.4% of total deletions, respectively (Fig. 3a, lower). Most NHEJ-based deletions were <20 bp, whereas MMEJ-based deletions showed a wider size range with an average of 9.5 bp microhomology shared at the breakpoints (Fig. 3b). To determine whether the observed dominance of NHEJ- and MMEJ-based deletions was due to variable performance of PhaseDel across different mechanisms, we examined the fraction of deletion candidates from each mechanism that were filtered out through the linkage analysis, a critical step in PhaseDel (Supplementary Fig. S4c–e). All mechanisms except for MMEJ and nonallelic homologous recombination (NAHR) demonstrated similar fractions of filtered events (Supplementary Fig. S4d). Sequence homology between the two breakpoints in MMEJ- and NAHR-based deletion candidates increases the chance of chimeric artifact formation during in vitro genome amplification likely resulting in the higher fractions of false positives filtered by the linkage analysis. Therefore, NHEJ- and MMEJ-dominance in our call set likely reflects the major mechanisms underlying somatic deletions in post-mitotic human neurons.

Post-mitotic neurons accumulate somatic SNVs with age, termed genosenium[10]. PhaseDel estimated the genome-wide rates of somatic nanodeletions and revealed a consistent accumulation with age

($P = 2.0 \times 10^{-8}$, linear regression; Fig. 3c, left). Nanodeletions accumulated at a rate of 1.7 per year, ~7.5% of the reported rate for sSNVs (22.6 sSNVs per year)[10]. Our observation of increasing deletion burden with age contrasts with a recent report of the negative correlation between age and the prevalence of neurons with larger somatic CNVs[14]; this inconsistency could be explained by the size difference of targeting variants (megabase-scale events in the previous study), exerting different selective pressure for cell survival. Somatic deletions were enriched in genes involved in neuronal function after accounting for their size (Supplementary Fig. S5), demonstrating similar enrichment as neuronal sSNVs[10,11]. We separately estimated the burden of NHEJ- and MMEJ-based deletions, and both types of deletions significantly increased with age ($P = 3.56 \times 10^{-5}$ and $P = 3.16 \times 10^{-4}$, respectively, linear regression; Fig. 3c). Overall, NHEJ-based deletions showed a two-fold greater rate of accumulation compared to MMEJ-based deletions (0.86 and 0.44 per year, respectively), although the contribution ratio between NHEJ and MMEJ was highly variable across the cells.

In post-mitotic single neurons, genes highly expressed in the brain showed an increased burden of sSNVs, suggesting an increase of transcription-coupled DNA damage[10,11]. Similar to sSNVs, somatic nanodeletions showed significant enrichment in highly expressed brain genes (see Methods), but only for NHEJ-based deletions, not for MMEJ deletions (empirical $P = 0.004$ and $P = 0.14$, respectively; Fig. 3d). We postulated that the burden of NHEJ-based deletions would increase with gene expression levels, and tested this by measuring the

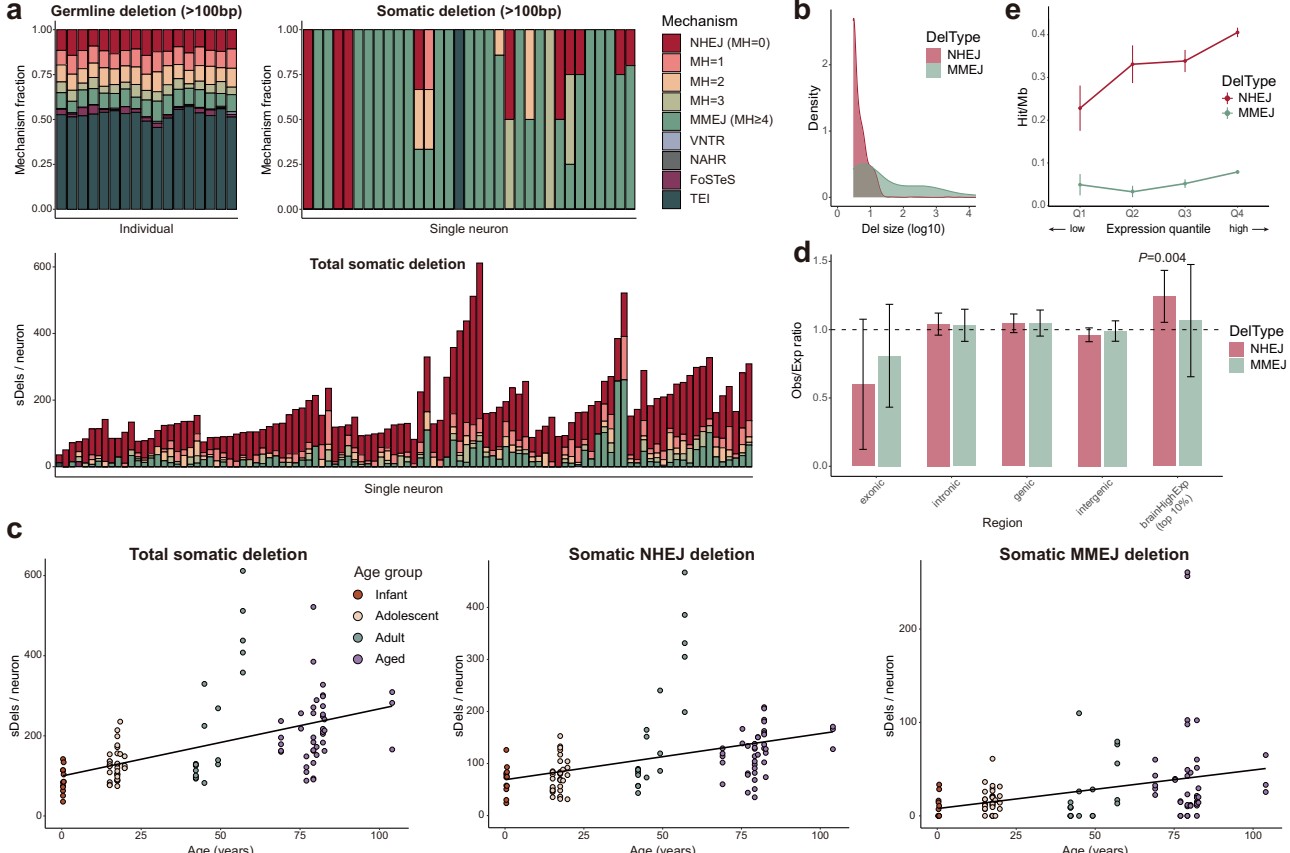

**Fig. 3 | Somatic nanodeletions in single-neurons increase with age and reflect distinctive underlying repair mechanisms. a** Contributions of the predicted mechanisms to deletion formation: germline deletions (>100 bp) per individual (upper left, 18 individuals), somatic deletions (>100 bp) per cell (upper right, 31 cells), and total somatic deletions per cell (lower, 107 cells). Individuals and single neurons are presented in order of age and deletion rate within each individual. Single cells with all deletions <100 bp are omitted in the upper right panel. Detailed criteria for mechanism prediction are described in Methods. MH microhomology. **b** The size of NHEJ- and MMEJ-based somatic deletions. **c** Somatic deletion counts by age with linear regression lines. Each point represents a single neuron. Total (left), NHEJ-based (MH = 0, middle), and MMEJ-based (MH ≥ 4, right) somatic deletions all showed a significant increase with age ($P = 3.3 \times 10^{-6}$, $4.78 \times 10^{-5}$, $1.44 \times 10^{-3}$; linear mixed model). **d** Enrichment of somatic deletions in different genomic regions. Y axis represents the ratio of somatic deletion counts for each category to expected counts from simulation. $n = 1000$ independently simulated sets; bar graph, mean ± 95% CI; empirical $P = 0.004$. **e** Somatic deletion burden by gene expression quantile in normal PFC. $n = 1000$ bootstrap deletion sets; mean ± SEM. Source data are provided as a Source Data file.

burden per quartile of gene expression level in normal PFC (see Methods). This analysis confirmed a positive correlation between deletion burden and gene expression, again only for NHEJ but not MMEJ, suggesting that NHEJ is the predominant mechanism to repair transcription-induced DSBs in human neurons (Fig. 3e).

## Somatic nanodeletions increase with distinctive patterns in neurons of individuals with DNA repair defects

Defective DNA repair caused by inherited mutations can lead to congenital neurodegenerative diseases[2,5,6]. AT is a rare genetic disease characterized by progressive neurodegeneration, cancer predisposition, and premature aging caused by recessive mutations in the *ataxia telangiectasia mutated (ATM)* gene[34,35]. ATM is a key protein kinase that transmits a DSB damage signal to cell-cycle-checkpoint proteins such as p53 and CHK2, arresting cell cycles for DSB resolution or triggering DSB-induced apoptosis[36,37] (Fig. 4a). ATM deficiency in AT has been thought to result in accumulation of DNA-damaged neurons, which may lead to cell loss and further neurodegeneration[38]. Although a high predisposition to cancer—predominantly leukemia and lymphoma[35,39]—and a high prevalence of chromothripsis in tumors of patients with AT[40,41] have been reported, direct assessment of DSB accumulation in human single neurons of AT patients has not yet been explored. Therefore, we generated MDA-based high-depth scWGS data (~45×) for 11 PFC neurons from two AT patients in this study.

AT single neurons demonstrated an excess of somatic nanodeletions compared to the neurons of age-matched normal controls ($P = 4.6 \times 10^{-3}$, two-sided Mann–Whitney $U$ test; Fig. 4b). The average estimated burden in single neurons of 19- and 24-year-old AT patients (185.6 per neuron) was at a similar level to the 51-year-old neurons in the control PFC, suggesting accelerated accumulation of DSB events in AT. Note that the sSNV rates estimated with LiRA, a similar phasing method for sSNV[23], showed no difference between neurons of AT and normal controls (Supplementary Fig. S6a), suggesting the role of ATM loss specific to DSB-induced deletions but not to sSNVs. The increase in somatic nanodeletions in AT was almost completely derived from NHEJ ($P = 9.35 \times 10^{-3}$, two-sided Mann–Whitney $U$ test; Fig. 4b), consistent with previous reports of the enhanced NHEJ burden in ATM-deficient replicating cells[42,43].

We have previously shown an increase in somatic SNVs in two genetic diseases with early-onset neurodegeneration and premature aging—CS and XP, both resulting from defective NER[10]. A recent comprehensive study of genotoxic exposures and DNA repair deficiencies in *Caenorhabditis elegans* found that defective NER not only results in an increase in sSNVs but also increases indels and SVs, especially deletions of 50–400 bp[44]. We thus examined somatic nanodeletion accumulation in CS and XP using previously published high-depth (~45×) MDA scWGS dataset of 26 and 13 PFC neurons of CS and XP patients, respectively[10].

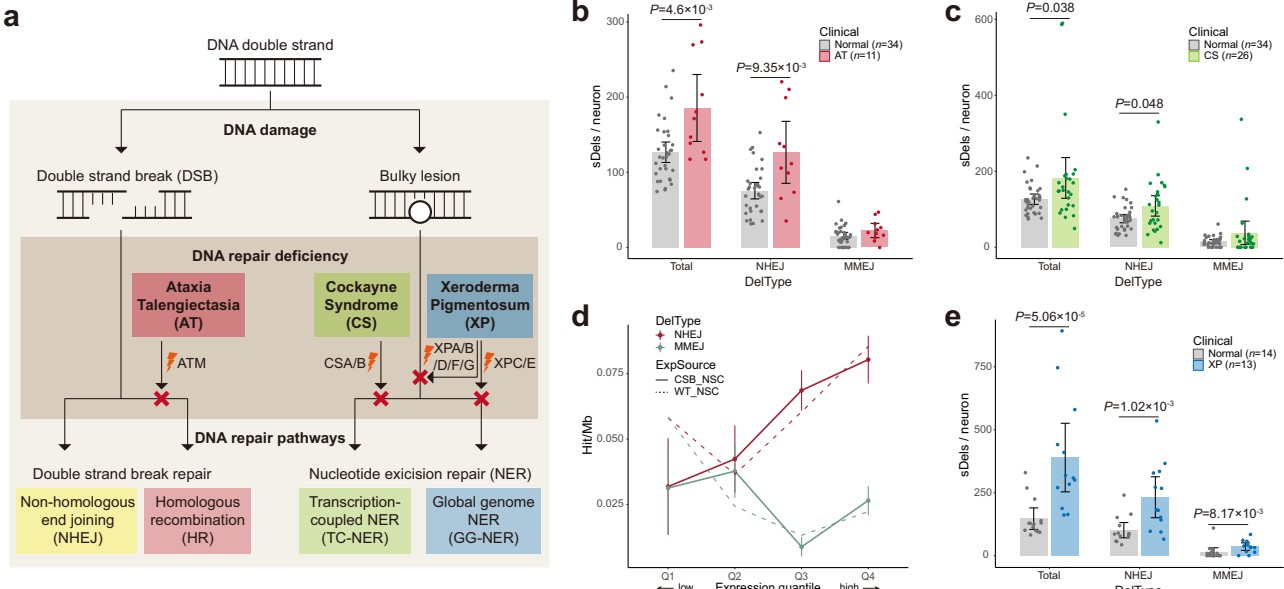

**Fig. 4 | Increased burden and distinctive patterns of somatic deletions in neurons of individuals with DNA repair deficiencies. a** Three DNA repair-defective diseases (AT, CS, XP) and their corresponding defective repair pathways. **b**, **c**, **e** Total and NHEJ- and MMEJ-based deletion burdens in AT (**b**), CS (**c**), and XP (**e**) single neurons compared to age-matched controls. *n*, number of single neurons; bar graph, mean ± 95% CI; two-sided Mann–Whitney *U* test. **d** Distinctive associations of somatic deletion burden in CS PFC neurons with gene expression in different conditions. RNA-seq data from iPSC-derived neural stem cells from CS patients (CSB_NSC, solid lines) and normal controls (WT_NSC, dashed lines) were used to associate with the burden of NHEJ- (red) and MMEJ-based deletions (green) from CS PFC neurons. *n* = 1000 bootstrap deletion sets; mean ± SEM. Source data are provided as a Source Data file.

We observed a considerable increase in somatic nanodeletions in CS neurons compared to age-matched controls ($P = 0.038$, two-sided Mann–Whitney *U* test; Fig. 4c), but only for NHEJ-based deletions ($P = 0.048$, two-sided Mann–Whitney *U* test). DNA repair defects in CS specifically involve transcription-coupled NER (TC-NER) (Fig. 4a), and we observed that NHEJ was the predominant pathway involved in repairing transcription-induced DSBs in neurons (Fig. 3e). Comparison of somatic nanodeletion rates to gene expression data from normal and CS-patient iPSC-derived neural stem cells (NSCs)[45] (see Methods) also revealed consistent results. The rates of NHEJ-based but not MMEJ-based nanodeletions from CS PFC neurons increased with gene expression in iPSC-NSCs from CS patients (Fig. 4d, solid lines), but not with gene expression in iPSC-NSCs from control individuals (Fig. 4d, dashed lines). In line with this, we found no relationship between the NHEJ burden in CS PFC neurons and gene expression in the normal PFC (Supplementary Fig. S6b).

There was a significant increase in the deletion burden in XP single neurons compared to age-matched normal controls ($P = 5.06 \times 10^{-5}$, two-sided Mann–Whitney *U* test; Fig. 4e). There are eight XP complementation groups according to mutated gene, and all but one of these fit into two major subgroups: one group (XPC/E) adversely affects only global genomic NER (GG-NER); the other group (XPA/B/D/F/G) adversely affects both TC- and GG-NER by disrupted repair factors that are commonly used for both downstream pathways[2,6] (Fig. 4a). Our XP samples are from the subgroup that affects both NER pathways (XPA/D); we expected the NER defects to be causative and therefore to observe a deletion burden caused by both NER defects. Notably, in XP neurons, there was a significant increase in deletions not only for NHEJ ($P = 1.02 \times 10^{-3}$, two-sided Mann–Whitney *U* test) but also for MMEJ ($P = 8.17 \times 10^{-3}$, two-sided Mann–Whitney *U* test), the latter suggesting a specific burden from defective GG-NER since CS neurons with only defective TC-NER did not show an increasing burden of MMEJ deletions (Fig. 4e). Despite the significant enrichment of MMEJ-based deletions, the overall increase in deletions in XP neurons was mainly derived from NHEJ deletions (~6.4-fold higher burden). This suggests

that defective TC-NER is a major contributor of deletions detected in the XP single neurons.

## Discussion

Recent scWGS analyses have successfully characterized somatic mutations in single cells, but have been limited to SNVs, transposable elements (TEs), and large CNVs[10–14,16,20,21,23,46]. In this study, we developed PhaseDel, a method that utilizes phasing information to delineate true somatic deletions from SV-like artifacts in scWGS data. We identified and validated somatic nanodeletions, focal deletions generally <1 kb in size, from whole-genome-amplified single-cell sequencing data. PhaseDel will also be useful for the detection of germline deletions, for example, those in complex regions such as human leukocyte antigen loci as the linkage between deletion-supporting reads and other adjacent germline variants will help reduce false positive predictions.

Our analysis of single post-mitotic neurons using PhaseDel revealed the accumulation of somatic nanodeletions with age, in line with previous reports of somatic SNV accumulation[10,12,47]. The accumulation rate of somatic nanodeletions was 13-fold less than that of sSNVs, and notably, the estimated rate of deletions per cell from any particular individual varied more than did the rate of sSNVs[10] (Supplementary Fig. S6d). The higher variability in rates of somatic deletions compared to sSNVs may reflect differences in selective pressures, since deletions are more likely to be damaging than SNVs by involving multiple bases and potentially causing frameshifts. Consistently, a recent study reported a negative correlation between age and the prevalence of neurons with somatic CNVs, which are megabase-scale in size and thus exert much stronger selective pressure[14]. In that study, while they observed anti-correlation between age and the fraction of neurons harboring CNVs, they also observed a higher average number of somatic CNVs in CNV-harboring-neurons from aged groups, likely supporting both the accumulation of CNVs and the increased selective pressure with age. The study also suggested that gene transcription is the main cause of neuronal CNVs by showing the significant

enrichment of somatic CNVs in long genes that are highly expressed and involved in neuronal functions[48–50]. These common findings between the previous study and our analyses further support the shared origin of different-sized deletions and their variable selective pressure.

Our analysis revealed an increased burden of somatic deletions in genes with high levels of expression, especially for NHEJ-based deletions. Transcriptional DSBs have been reported in post-mitotic neurons[51,52], but more investigation is needed to determine why there is a positive correlation between NHEJ-based deletions and gene expression levels, but not for MMEJ-based deletions. One possible explanation is that transcription involves multiprotein complexes containing NHEJ proteins and RNA polymerase II (RNAP II), resulting in enhanced recruitment of NHEJ proteins to highly transcribed genes[53]. For nascent RNA-templated DSB repair in post-mitotic neurons, multiprotein complexes with both RNAP II and DSB repair proteins have been reported to be preferentially accumulated at DSB sites of transcribed genes[53,54]. We speculate that the recruited NHEJ proteins are not only involved in RNA-mediated repair but also involved in classical NHEJ repair, resulting in an increased NHEJ efficiency. The absence of MMEJ proteins within the RNAP II complex further supports this possibility[53].

Researchers have long suspected that inactive ATM in AT neurons would result in the accumulation of somatic deletions derived from unrepaired DSBs, but no detection methodology for single-neuron WGS has been available to test this hypothesis. Our analysis using PhaseDel has confirmed this long-standing hypothesis. An increase in somatic deletions was also observed in two additional genetic diseases, CS and XP, that involve defective NER rather than defective DSB repair. Not all but many CS and XP cases present with neurodegenerative phenotypes similar to AT. For example, XP patients with inactivation of the *XPG* gene showed a loss of cerebellar Purkinje cells[55], and mice with dual inactivation of CSB and XPA genes demonstrated cerebellar atrophy with granule cell loss[56]. Since DSB accumulation is a direct cause of cell apoptosis or cell death, we propose that the accumulation of DSBs may contribute to neurodegeneration in CS and XP. We observed that the majority of increased DSBs in CS and XP neurons were derived from defective TC-NER, not from defective GG-NER. Consistently, XP patients without defective TC-NER are known to have little neurological impairment[6,57]. Taken together, DSB accumulation may serve as a common underlying mechanism of neurodegeneration.

Although our method can estimate a genome-wide deletion burden, it can directly detect events only in ~25% of the genome due to the requirement of nearby germline heterozygous SNPs. Thus, our analysis has limitation in characterizing the enrichment of deletions particularly in small genomic regions. For example, a previous study reported neuronal-activity-induced DSBs within the promoters of a few early-response genes[4], but these extremely small promoter regions were largely under-powered in our phasing analysis. Future improvements in genome amplification and the application of long-read sequencing will greatly improve PhaseDel's power, allowing further insights into the implications of DSBs in human neurons.

## Methods

### Human data and specimens

scWGS data for 107, 26, and 13 PFC single neurons, from post-mortem brain tissue of 18 neurotypical individuals, six CS, and three XP patients respectively, were obtained from previously published work[10–12]. Additional scWGS data for 11 PFC single neurons from two individuals with AT were generated. Post-mortem brain tissue was obtained from two female patients with AT, ages 19 and 24 years, from the NIH Neurobiobank. These samples were collected and processed according to a standardized protocol (Protocol Method 2, https://www.medschool.umaryland.edu/btbank/Researchers/Tissues-Collected/)

and under the supervision and approval of the NIH Neurobiobank and the Boston Children's Hospital institutional review board. Descriptive details of all individuals, such as age, sex, clinical status, and number of cells, are available in Supplementary Table S1.

### Single-cell isolation and sequencing

Bulk DNA was extracted using the buffer ATL from QIAamp DNA Mini kit and proteinase K digestion. The isolation of single nuclei using flow cytometry after staining for the pan-neuronal marker, NeuN (Millipore MAB337X), and their WGA using MDA were described previously[10–12]. Briefly, nuclei were prepared from fresh frozen human brain tissue samples, previously stored at −80 °C, in a Dounce homogenizer using a chilled nuclear lysis buffer (10 mM Tris-HCl, 0.32 M Sucrose, 3 mM Mg(Acetate)$_2$, 5 mM CaCl$_2$, 0.1 mM EDTA, pH 8, 1 mM DTT, 0.1% Triton X-100) on ice. Tissue lysates were layered on top of a sucrose cushion buffer (1.8 M Sucrose, 3 mM Mg(Acetate)$_2$, 10 mM Tris-HCl, pH 8, 1 mM DTT) and ultra-centrifuged for 1–2 h at 30,000 G. Supernatant was discarded and nuclear pellets were resuspended in ice-cold PBS supplemented with 3 mM MgCl$_2$, filtered, then stained with an anti-NeuN antibody in PBS supplemented with 3 mM MgCl$_2$ and 3% BSA (Millipore, MAB377X, clone A60, AlexaFluor-488 conjugated, 1:1250). Nuclei were then sorted by flow cytometry, one nucleus per well into 96-well plates with each well containing 2.8 μl alkaline nuclear lysis buffer (200 mM KOH, 5 mM EDTA, 40 mM DTT) prechilled on ice. Nuclei were lysed on ice for 15–30 min, and then neutralized on ice in 1.4 μl neutralization buffer (400 mM HCl, 600 mM Tris-HCl, pH 7.5). Then, MDA was performed. New AT samples were MDA amplified using the following protocol. 2 μl 10× Phi29 reaction buffer (Epicentre), 8.4 μl nuclease-free water, 4 μl 10 mM dNTP, 1 μl 1 mM random hexamer (5′ dNdNdNdN*dN*dN-3′ [where * is thiophosphate linkage]) (IDT or ThermoFisher), 0.4 μl repliPHI polymerase (40U) (Epicentre)) was added to each well. MDA was performed at 30 °C for 16 h then Phi29 was inactivated at 65 °C for 3 min. Supplementary Table S1 lists which samples were subject to Epicentre or Qiagen MDA.

Amplified single-cell genomes were subjected to quality control by DNA quantitation (picogreen, 3 μg yield required) and multiplex PCR for four random genomic loci. For germline reference, bulk DNA was purified using phenol:chloroform:isoamyl alcohol extraction and isopropanol precipitation, without RNAse A treatment. Amplified single-neuron genomes were prepared for sequencing by DNA shearing and libraries generated by Psomagen (Macrogen) using Illumina TruSeq kits and Illumina HiSeq X10 paired-end sequencing (150 bp × 2), as described previously[10].

### Read mapping and quality-control analysis

Sequenced reads were aligned to the human reference genome (GRCh37) using BWA-mem (version 0.7.17)[58] with default parameter settings. For each mapped BAM file, duplicated reads were marked using Picard (version 2.8.0), and indel realignment and base-quality recalibration were performed using the Genome Analysis Toolkit (GATK, version 3.5)[59].

As a quality control, we checked the evenness of WGA to exclude cells with failed amplification. Specifically, for each BAM file, the evenness of genome amplification was measured using the median absolute pairwise difference (MAPD) metric as previously reported[13]. The MAPD metric was calculated by taking the median value across all absolute differences between the copy number ratio of neighboring bins, which were divided to have the same number of uniquely mapped reads. Previous single-cell copy number analysis protocol[60] was applied to calculate MAPD values, including binning, GC normalization, segmentation, and copy number estimation. Single cells with MAPD > 2 were considered to result from failed WGA and excluded from the downstream analysis. Fifteen single cells that failed to estimate the deletion rate using PhaseDel were also excluded (see deletion rate estimation section below). scWGS data from a total of 107 and 50

PFC single-neurons from 18 normal individuals and 11 neurodegenerative disease patients were finally included in the study. Detailed sample information is described in Supplementary Table S1.

## Variant calling for linkage analysis

Germline heterozygous SNPs were identified from bulk sequencing data by GATK with default parameter settings. To include only high-confidence germline heterozygous SNPs for linkage analysis, we selected and used only the heterozygous SNPs (0/1 genotype) that were annotated as known SNPs in the dbSNP database (version 147)[61]. In a similar way, germline heterozygous deletions were identified from bulk sequencing data by GATK and DELLY2[25] with the default parameter settings. We selected putative germline heterozygous deletions (0/1 genotype from either GATK or DELLY2) if >90% of a deletion region overlapped with >90% of a known polymorphic deletion region in the dbSNP (version 147) or 1000 Genomes Phase 3 SV database (estd219)[62], allowing up to a 2- or 10-bp difference in deletion breakpoints for GATK and DELLY2 calls, respectively. Pairs consisting of a germline heterozygous SNP and a germline heterozygous deletion that were linked by at least two spanning reads (deletion-supporting clipped/discordant reads crossing the SNP site) were utilized as an answer set to measure the sensitivity of the PhaseDel algorithm with single-cell sequencing data.

Variant calling of initial somatic deletion candidates was performed by GATK and DELLY2 for single-cell sequencing data, just as was done for germline deletions. Called deletion candidates that were linked to germline heterozygous SNPs with at least two spanning reads (called phaseable candidates) were subjected to linkage analysis to further clarify whether they were true somatic deletions or chimeric artifacts occurring during single-cell WGA. All phaseable candidates were analyzed using linkage analysis including low-quality calls with low variant scores or filtered calls annotated with FILTER tags from GATK/DELLY outputs to maximize sensitivity. Initial phaseable call sets from GATK and DELLY were merged to make an integrated call set for every single cell. Deletions with call sets in both callers with an overlap of >90% were considered to be the same deletion and thus merged into a single event.

## Linkage analysis to detect somatic deletions

Each phaseable deletion candidate was reanalyzed to refine the exact position of the deletion breakpoint linked to the germline heterozygous SNP. Shared clipped positions of deletion-supporting reads were used as predicted breakpoints for linkage analysis, and the consensus sequence for the clipped-out part was constructed for each position based on reads with the majority of clipped subsequences. Phaseable reads (i.e., reads spanning the deletion breakpoint and the germline heterozygous SNP site) were then tagged as deletion-supporting or non-supporting reads for further linkage analysis. A given phaseable read was considered to be deletion-supporting if one of the following conditions is satisfied: (1) read clipping occurred within 3 bp of the predicted breakpoint and the clipped-out part matched the consensus sequence for the given breakpoint with >70% concordance, (2) read clipping did not occur and the read subsequence beyond the predicted breakpoint failed to match the reference sequence but rather matched the consensus sequence with >70% concordance, or (3) it was a discordant read (insert size > mean + 3 standard deviation of insert size for a given cell) and a given read and its mate in the pair were both mapped on opposite ends outside of the predicted deletion region.

For each phaseable heterozygous SNP site, the number of deletion-supporting/non-supporting reads was counted for each SNP allele. The allele with the larger number of deletion-supporting reads was designated to represent the deletion-containing haplotype. A given deletion was considered to be a false positive WGA artifact if (1) more than one of the phaseable reads with the deletion-supporting SNP allele were non-supporting reads or (2) at least one of the phaseable reads with the non-supporting SNP allele was a deletion-supporting read. When a given deletion candidate was linked to multiple heterozygous SNPs, it was filtered out as a false positive if any single SNP satisfied one of those two criteria. Additional filtering steps were also applied to remove more false positives based on the previously reported characteristics of WGA artifacts[22]. Candidates were filtered out if (1) >40% of phaseable deletion-supporting reads were inverted (F1F2 or R1R2) since inverted amplification is a major characteristic of a chimeric artifact[22], or (2) an average of the minor allele frequency (MAF) for all germline heterozygous SNPs located within the deleted regions was >10% since the MAF is theoretically zero for the SNPs within true somatic deletion regions identified in single-cell sequencing data.

True somatic and germline deletions can both survive the linkage analysis and additional filtering steps. To classify germline deletions, phased deletion regions were analyzed from bulk sequencing data from the same individual. A given candidate was classified as a germline deletion if the matched bulk data had deletion-supporting reads and if they were properly phased with the deletion-supporting SNP allele (with up to one non-supporting read along with the deletion-supporting allele) for all phaseable deletion breakpoints. Sensitivity was measured by comparing these phased germline deletions to the answer set for benchmarking. For remaining phased deletions, possible germline candidates were additionally filtered out if (1) there were no phaseable reads spanning the deletion-supporting SNP allele and the breakpoint in the bulk data, making it impossible to determine whether it was a germline deletion, or if (2) among the reads with the deletion-supporting SNP allele in the bulk data, the fraction of deletion-supporting reads was too high (>80%). Candidates that survived this filtering step were considered to be final somatic deletion candidates.

## Measuring sensitivity of PhaseDel

Putative germline heterozygous deletions annotated as known polymorphic deletions by public databases served as a gold-standard answer set. A given gold-standard germline deletion can be missed by PhaseDel from scWGS data if (1) there is no nearby germline heterozygous SNP, making linkage analysis inapplicable, (2) genomic regions around the deletion are not covered in scWGS data due to allelic/locus dropout during the WGA process, or (3) the variant was missed/filtered out by PhaseDel (false negative). In order to examine the performance of PhaseDel, which is only in play in the third condition, we examined each cell and divided the number of identified gold-standard germline deletions by the total number of phaseable gold-standard germline deletions, that is, deletions with at least two spanning reads in a given scWGS data, to which linkage analysis applicable. All calculated fractions were averaged and shown with a 95% confidence interval in Fig. 1e. To determine the fraction of total deletions expected to be identified by PhaseDel, we further calculated the number of identified deletions over the total gold-standard germline deletions including in non-phaseable regions in a given cell (Supplementary Fig. S1f). The total number of gold-standard germline deletions, the number of phaseable and identified deletions by PhaseDel, and all measured fractions for each cell are described in Supplementary Data 1.

## Estimation of somatic deletion rate

PhaseDel can only detect somatic deletions that are close to germline SNPs (covering ~25% of the entire genome, Supplementary Fig. S1f); therefore, extrapolation is required to estimate the genome-wide somatic deletion rate. In addition, although final somatic candidates were phased and filtered to remove false positives, such false calls might still have remained if the candidate region was under-powered due to uneven amplification or locus/allelic dropout, in which case the region with low read depth would have been more likely to have a

random phased false call or to have just one linkage-violating read. To accurately estimate the genome-wide rate while controlling false positives, we developed a two-component model that separately estimates underlying rates of true somatic deletions and false positive errors for sequencing data for a given single cell, which is a strategy similar to that adopted in our previous somatic SNV analysis[23].

The main idea was to analyze all phased candidates, categorize them according to the number of candidate-supporting linked-read count, calculate the rate at which candidates with the same linked-read count occurred compared to all genomic positions with that linked-read count, and then separate them into the rate for true somatic deletions and for errors based on expected differences in the rate of each (Supplementary Fig. S1g). For true somatic deletions, we assumed there to be no difference in the expected rate of true deletions according to their deletion-supporting linked-read counts, since amplification bias randomly affects the entire genome, and thus there is no reason to expect a higher rate of somatic deletion with certain supporting read counts. However, for false positive errors, we assumed that higher supporting read counts would yield fewer errors, that is, regions with higher read-depth are less likely to have random errors that are perfectly linked (phased) or with just one linkage-violating read. Therefore, the rate of all phased candidates ($T(c)$) with supporting read counts ($c$) was considered to be a mixture of a constant and a decaying function, representing the rates of true somatic deletions ($S(c)$) and false positives ($E(c)$), respectively (Supplementary Fig. S1g).

$$T(c) = S(c) + E(c) = k_1 + k_2 e^{-k_3(c-2)}; c \geq 2, k_1 \geq 0, k_2 \geq 0, k_3 \geq 0 \quad (1)$$

The exponential decay model was selected to fit the decaying function for the rate of errors based on the measured pattern of the rate of all phased candidates for each supporting linked-read count (Supplementary Fig. S1g). At least two phaseable reads were required for linkage analysis, so the minimum count for supporting reads was set to two ($c \geq 2$). Parameters $k_1$, $k_2$, and $k_3$ were estimated by Bayesian inference using the Markov chain Monte Carlo (MCMC) sampling, conducted using the R package rstan. Four MCMC chains were used, each with 5000 burn-in steps followed by 10,000 iterations.

For a given supporting read count ($c'$), the total rate ($T(c')$) was calculated as the number of candidates with $c'$ supporting reads divided by the total number of positions covered by $c'$ phaseable (linked) reads. For all phaseable positions in a given cell, we counted the number of positions per phaseable read count to calculate the rate for each supporting read count. When a given position was linked to multiple germline heterozygous SNPs and thus had multiple phaseable read counts, then the highest read count was assigned as the representative count for the given position. Calculated rates across all phaseable read counts (black dots, Supplementary Fig. S1g) were used to fit the mixture model of two components, true somatic deletions, and errors (blue horizontal line and orange curve, Supplementary Fig. S1g), as described above. The posterior mean and 95% confidence interval of $k_1$ were reported as an estimated somatic deletion rate for a given cell. If the model failed to converge or estimated parameters violated the constraints for a given cell (e.g., the rate for all phased candidates did not decay as supporting read counts increased, but rather that rate increased due to severe amplification bias), then it was considered to be a cell with excessive amplification errors and was excluded from the study. A total of 157 single cells were used in downstream analyses after excluding 15 single cells (13, 1, and 1 from normal, CS, and XP cells, respectively) with failed estimation.

Based on the fitted two-component model, the FDR for a given supporting read count ($c'$) was estimated as follows: $FDR(c') = E(c') / T(c')$. Utilizing this FDR estimation, PhaseDel constructed high-confidence candidate sets that were controlled to have a similar FDR (<10%) across all cells by setting a cell-specific supporting read count threshold. With the read count threshold ($c_t$), the aggregated FDR for a given cell was calculated as follows:

$$FDR_{aggr}(c_t) = \frac{\sum_{c \geq c_t} E(c)}{\sum_{c \geq c_t} T(c)} \quad (2)$$

For each cell, the minimum threshold ($c_{t,min}$) to satisfy $FDR_{aggr} < 0.1$ was determined and a high-confidence somatic deletion set was constructed by selecting phased deletion candidates with supporting read counts $\geq c_{t,min}$. These FDR-controlled high-confidence sets were used for the entire analysis in this study. Estimated somatic mutation rates, supporting read count thresholds, and the number of selected high-confidence somatic deletions for each cell are described in Supplementary Data 1.

## Evaluation of PhaseDel accuracy using the published kindred clone WGS data

Previously published genome sequencing data of single-fibroblast-derived clones[27] were used to assess the performance of PhaseDel. Kindred-cell-specific somatic deletions (Supplementary Fig. S3a, red stars) served as a gold-standard set for comparison: somatic deletion candidates from the two MDA-amplified scWGS datasets (IL-11, IL-12) and unamplified bulk WGS data of the kindred clone (IL-1c) were compared to each other. Specifically, we evaluated (1) what fraction of PhaseDel deletion calls from scWGS are observed in the kindred clone WGS, and (2) whether the genome-wide deletion rate estimated from scWGS by PhaseDel is comparable with the actual number of somatic deletions observed in the clone data.

To confirm the presence of a scWGS-derived PhaseDel deletion in the kindred clone, we checked supporting reads for the corresponding deletion from the bulk clone WGS data. A read was considered deletion-supporting only if it has both the same clipping/indel position and the clipped/indel sequence as the supporting reads from the original scWGS data. A single-cell deletion event was considered confirmed if it has ≥5 total reads spanning the breakpoint and ≥3 deletion-supporting reads in the bulk clone WGS data.

To assess the reliability of PhaseDel's deleteion rate estimation, we presumed the actual number of kindred-cell-specific somatic deletions by counting the number of deletions from the clone WGS data that are not observed in any other data except the kindred groups. Multiple filtering steps were applied to make conservative deletion call sets (Fig. S3c). Specifically, candidates were filtered out if (1) their breakpoints matched with known polymorphic deletions reported in the 1000 Genomes structural variation database[63] or the Database of Genomic Variants[64], (2) more than one deletion-supporting read were observed in the matched bulk WGS, or (3) there were more than one deletion-supporting read in any bulk WGS data of unrelated individuals from our study or in any scWGS data from the non-kindred group of Dong et al.[27] and our study. Deletion-supporting reads were identified by checking both the clipping position and their clipped sequence as described above. The scWGS-derived deletions rates by PhaseDel were then compared to this count from the kindred clone WGS to assess their consistency.

## Prediction of underlying DSB repair mechanism

Underlying repair mechanisms for deletion candidates were predicted following criteria in a previous study[31]. For each deletion candidate, the genomic element and sequence homology at the deletion breakpoint were analyzed to classify the resulting deletions into six different categories: TEI, variable number of tandem repeat (VNTR), NAHR, FoSTeS/microhomology-mediated break-induced repair (FoSTeS/MMBIR), NHEJ, and MMEJ (Supplementary Fig. S2b). The gnomAD database[65] and RepeatMasker information obtained from the UCSC genome browser were utilized to annotate known TEs and repeat regions. A given deletion was classified as a TEI event if >80% of a

deletion region and >80% of a known TE overlapped. A given deletion was classified as a VNTR event if >80% of a deletion region overlapped with >80% of a known repetitive element from simple repeats (microsatellites), satellite repeats, or low complexity repeats. A given deletion was classified as a FoSTeS/MMBIR event if either of the two deletion breakpoints had a >10 bp insertion. The other three types were determined based on the length of sequence homology between the deletion breakpoints. If a given deletion had >100 bp sequence homology around the two breakpoints, it was classified as an NAHR event. If the homology length was from 4 to 100 bp, it was classified as an MMEJ event. If a given deletion lacked sequence homology between the two breakpoints, it was classified as an NHEJ event. Note that deletion candidates with sequence homology of 1–3 bp might originate from both NHEJ and MMEJ, so they were treated as an independent group (MH = 1, 2, 3) and were excluded from the entire NHEJ/MMEJ-based analysis.

### Validation sequencing of somatic deletions

Initially, a total of 66 somatic deletion candidates were randomly selected for validation sequencing and tested in two batches, the first targeting 35 candidates and the second targeting 38 candidates, with seven candidates targeted in both batches to double-check the validity of the sequencing approach. We additionally selected 197 more candidates to include candidates from all 29 individuals and diverse mechanisms. Of the total 263 candidates, 19 failed to generate any reads even for the wild-type alleles without the deletions. Exclusion of these 19 candidates from the validation list resulted in 244 deletions with no candidates for two normal individuals (5559 and 5943).

Custom amplicons were designed to cover each deletion breakpoint to directly capture DNA fragments spanning a breakpoint. Two different primer pairs were designed for each deletion candidate, generating two different amplicons. In general, the primer pairs were designed to have amplicons smaller than 500 bp in size when a deletion was present, and when this was not possible, the primers were designed to have the deletion junction within a single read size from at least one side of the amplicon. After designing primers to candidate loci, common handle sequences were added to each to allow for sample-specific index sequences to be added to pooled amplicons, allowing for multiplex sequencing in single runs. The structure of the primers used to generate each amplicon was 5′-ACACTCTTTCCCTACA CGACGCTCTTCCGATCT-[GENE SPECIFIC SEQUENCE]-3′ and 5′-[GENE SPECIFIC SEQUENCE] AGATCGGAAGAGCACACGTCTGAACTCCAGTC AC-3′. Indexes were added in a subsequent PCR using 5′-AATGAT ACGGCGACCACCGAGATCTACACTCTTTCCCTACACGACGCTCTT-3′ and 5′- ACACGTCTGAACTCCAGTCACNNNNNNNATCTCGTATGCCGTC TTCTGCTTG-3′, where NNNNNNN represents a 7nt index sequence that was unique to each sample. PCR cycles were minimized in both amplification steps to reduce amplification artifacts.

PCRs using the designed primers were performed using Platinum Taq (ThermoFisher/Life Technologies 10966083) and purified using magnetic beads (Ampure) on four different DNA sample types: WGA DNA from the deletion-called single cell, WGA DNA from an uncalled negative control cell, bulk PFC DNA from the deletion-called individual, and bulk non-brain DNA from the deletion-called individual. The amplicons were indexed and pooled, and then sequenced on MiSeq with read lengths of 2 × 250 bp (first batch) and 2 × 300 bp (second and third batch), targeting an average depth over 50,000×.

Sequenced reads were aligned to the human reference genome (GRCh37) using BWA-mem and generated a single BAM file. Indel realignment and base-quality recalibration were performed on the BAM file using the GATK. The BAM file was split into four BAM files for each DNA source by demultiplexing reads based on the index information. For each source-specific BAM file, the number of deletion-supporting reads was counted for each candidate region. A given read was considered to be a deletion-supporting read if (1) read clipping

occurred at the predicted deletion junction and (2) clipped read sequences matched the predicted spanning sequences with <10% mismatches up to a maximum of five mismatched bases. A given deletion candidate was considered to be validated if two conditions were met. The first required condition was for the data generated from WGA DNA of a deletion-called single cell to show more than 10% of reads spanning the predicted junction to be deletion-supporting reads. Although the fraction of deletion-harboring genome in a single cell should be 50% for an actual deletion event, single-cell WGA generally results in high allelic imbalance and hence the cutoff of 10% was chosen. The second required condition was for <5 deletion-supporting reads to be identified from the WGA DNA of a negative control cell. This cutoff was chosen to allow unexpected supporting reads caused by index hopping; however, most validated candidates (80.9%) had no deletion-supporting reads in a negative control cell.

### Gene ontology enrichment analysis

Somatic deletions disrupting genic regions were selected and subjected to gene ontology (GO) enrichment analysis. Since larger genes are more likely to be affected by random deletions, we utilized the GREAT tool[66] implemented in R (rGREAT) to perform a binomial test over genic regions. This test measures the area of overlapping regions between deletions and genes rather than relying on simple counts of overlapping genes, thereby controlling bias caused by gene and deletion size. $P$ values were adjusted with FDR to correct for multiple testing. Significant GO terms (FDR-adjusted $P$ value <0.05) involved in neuronal functions were reported in Supplementary Fig. S5. Full list of enriched GO terms is described in Supplementary Data 2.

### Regression analysis of somatic deletion and age

The association between the estimated somatic deletion rate and age in normal PFCs was tested using a linear regression model, controlling for gender using R software (version 4.0.1). For each cell, the number of observed NHEJ- and MMEJ-based deletions over the number of total observed deletions in that cell was multiplied by the somatic deletion rate to estimate the rate of NHEJ and MMEJ-based deletions in the cell. Further regression analysis was performed to determine the association between each of the NHEJ and MMEJ deletion rates and age.

### Regional burden analysis of somatic deletions

The burden of somatic deletions in normal PFCs was measured to compare four types of genomic loci: intergenic, genic, exonic, and intronic regions. To check the burden of transcription-coupled damage, we also sought to compare these to the most highly expressed genes (top 10%) in normal PFCs. To define the top 10% highly expressed genes in normal PFCs, we obtained gene expression data from normal PFCs from the GTEx database[67] and measured gene expression levels controlling for age and gender using DESeq2[68]. An average of the expression levels for each gene across all the samples was calculated to make a single list of gene expression, and the genes in the top 10% were subjected to burden analysis. ANNOVAR[69] was used to annotate the genomic region and the affected genes for somatic deletion candidates.

The count of somatic deletions can be confounded by per-cell differences in allelic/locus dropouts occurring during WGA and in covered phaseable regions. To correct these biases, we simulated 1000 sets of random somatic deletions for each cell, generating the same number and the same size deletions from the original call but randomly distributed within the amplified and phaseable regions for each cell. We merged deletions from all cells into one set of deletions and measured the deletion burden in each of five regions, repeated this 1000 times with the simulated data, and calculated the expected burden. We then calculated an observed/expected ratio with the 95% confidence interval for each region, as shown in Fig. 3d. The ratio was compared to the expected distribution to obtain the empirical $P$ value.

Both the simulation and the burden test were separately performed for NHEJ- and MMEJ-based deletions to independently test the enrichment of each underlying mechanism.

## Comparison of normal vs diseased single neurons

The somatic deletion burden was calculated for AT, CS, and XP single neurons by taking the average of the estimated somatic deletion rates for each disease group, reported with 95% confidence interval. NHEJ and MMEJ deletion burdens were measured in the same way. The burdens were compared to those from age-matched normal neurons; 11 AT and 26 CS neurons were compared to 34 normal neurons from adolescent individuals (15–20 years) and 13 XP neurons were compared to 14 normal neurons from adult individuals (40–50 years). Pairwise group comparisons were performed using the two-sided Mann–Whitney $U$ test. Since the ages of samples within the disease and normal groups were not perfectly matched, we also compared the burden after correcting for age, by subtracting the predicted age-associated burden per cell using the estimated regression coefficient for normal cells. Supplementary Fig. S6c demonstrates that age-corrected deletion burdens were consistent with non-age-corrected burdens for all diseases and deletion types (Fig. 4b, c, e).

## Deletion burden analysis with gene expression

We obtained gene expression data on normal PFCs from the GTEx database and constructed a single list of all genes with average expression levels, as described before. The total gene list was then divided into four quartiles based on expression levels. Since there is no publicly available gene expression data directly measured from the brains of CS patients, we obtained instead gene expression data from iPSC-derived NSCs originating from the fibroblasts of CS patients harboring germline mutations in the *CSB* gene[45]. Per-gene read counts measured by HTSeq[70] were directly obtained from the GEO database (GSE124208) and normalized to RPKM (reads per kb per million reads). CS gene expression levels were then averaged across the samples and the total genes were divided into four quartiles as with normal PFCs. Gene expression data of iPSC-derived NSCs originating from wild-type fibroblasts from the same study were also analyzed in the same way to further verify whether the association with gene expression was derived from the *CSB* mutation and not from the characteristic of the iPSC-derived cell line itself.

For each clinical condition (e.g., normal, CS), we constructed a merged set of somatic deletions identified from all cells of the corresponding individuals. The density of somatic deletions (deletion hit/Mb) was calculated for the gene set in each quartile and depicted in a line graph showing all quartiles (Figs. 3e and 4d). To calculate the repair-mechanism-specific burden (NHEJ and MMEJ), deletion sets were divided into NHEJ and MMEJ sets based on their predicted mechanisms, and the density of each set was measured. The standard error in the somatic deletion burden was assessed by bootstrapping, which sampled the same number of deletions with replacement from the entire deletion set and calculated the density for 1000 times.

## Statistical analysis

All the statistical tests were performed using R software (version 4.0.1). Detailed information for statistical tests including the type of test for each analysis and number of cells/samples are described in Methods and figure legends. All information related to the figures including used measures, definition of error bars, definition and exact value of $n$, and statistical significance were described in figure legends. Statistical significance was defined as $P < 0.05$.

## Reporting summary

Further information on research design is available in the Nature Research Reporting Summary linked to this article.

## Data availability

Single-neuron whole-genome sequencing data of control individuals and of individuals with CS and XP from the published work have been deposited in the NCBI Sequence Read Archive (SRA) (SRP041470 and SRP061939), the NIH Alzheimer's disease genomic data repository (NIAGADS and NG00121), and dbGaP (phs001485.v1.p1 [https://www.ncbi.nlm.nih.gov/projects/gap/cgi-bin/study.cgi?study_id=phs001485.v2.p1]). Single-neuron whole-genome sequencing data of AT patients and targeted amplicon sequencing data for validation of selected deletion candidates are deposited in dbGaP (phs003005.v1.p1 [https://www.ncbi.nlm.nih.gov/projects/gap/cgi-bin/study.cgi?study_id=phs003005.v1.p1]). The NIAGADS and dbGaP data are available under controlled-use conditions with the data use limitations and the instructions for applying to access the sequencing data. Whole-genome sequencing data of single fibroblasts and single-fibroblast-derived clones used for method validation are obtained from the SRA (SRP067062). Gene expression data on normal PFCs and iPSC-derived NSCs are obtained from the GTEx database (V8 gene transcripts per million (TPM) data, https://www.gtexportal.org/home/datasets) and the GEO database (GSE124208). All data needed to evaluate the conclusions in the paper are present in the paper and/or the Supplementary information. Source Data are provided with this paper.

## Code availability

The implemented program with the source code and a user manual is available at https://sourceforge.net/projects/phasedel/.

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

## Acknowledgements
Human tissue was obtained from the NIH Neurobiobank at the University of Maryland Brain and Tissue Bank, and we thank the donors and families for their invaluable contributions to the advancement of science. We also thank Lixing Yang for helpful discussions regarding the prediction of underlying DSB repair mechanisms and Allie Tolles for assistance with validation sequencing experiments. This work was supported by NIH K01 AG051791 (E.A.L.); DP2 G072437 (E.A.L.); the ATCP foundation (E.A.L.); the Suh Kyungbae Science Foundation (E.A.L.); NIMH U01 MH106883 (C.A.W.); NINDS R01 NS032457 (C.A.W.); the Allen Discovery Center program, a Paul G. Allen Frontiers Group advised program of the Paul G. Allen Family Foundation (C.A.W. and E.A.L.); the National Research Foundation of KOREA (NRF) NRF-2022R1C1C1010430 (J.K.); Basic Science Research Program of the Ministry of Education (MOE, South Korea) NRF-2019R1A6A1A10073079 (J.K.); DOD W18XWH2010028 (J.K., E.A.L., and C.A.W.); the NIH T32 HL007627 (M.B.M.); K08 AG065502 (M.B.M.); R00 AG054748 (M.A.L.); the Brigham and Women's Hospital Program for Interdisciplinary Neuroscience through a gift from L. and T. Rand (M.B.M.); the donors of the Alzheimer's Disease Research program of the BrightFocus Foundation A20201292F (M.B.M.); and the Doris Duke Charitable Foundation Clinical Scientist Development Award 2021183 (M.B.M.). C.A.W. is an Investigator at the Howard Hughes Medical Institute.

## Author contributions
J.K., M.A.L, and E.A.L. conceived and designed the study. J.K. developed and implemented the PhaseDel algorithm. J.K., A.Y.H., and J.L. performed bioinformatic analyses. M.A.L. and M.B.M. performed single-neuron sorting and sequencing. M.A.L., S.L.J, L.I., and A.M.J. designed and performed custom-amplicon-based validation sequencing. C.A.W. and E.A.L supervised this study. J.K., M.A.L., and E.A.L. wrote the manuscript. All authors reviewed and edited the manuscript.

## Competing interests
The authors declare no competing interests.
