## [Peer Review File · Nature Communications]

Prevalence and mechanisms of somatic deletions in single human neurons during normal aging and in DNA repair disordersREVIEWER COMMENTS

Reviewer #1 (Remarks to the Author):

Title: Prevalence and mechanisms of somatic deletions in single human neurons during normal aging and in DNA repair disorders

In this study, the authors developed the PhaseDel algorithm, which utilized GATK and DELLY2 to call somatic deletion candidates, then filtered and read-back phased, using clipped reads spanning reliable germline heterozygous SNP sites called from bulk WGS data. They then tried to identify true somatic deletions from 108 single neurons, which increased with age and in highly expressed genes in the human brain. Also, they showed that somatic deletions increased with distinctive patterns in neurons of individuals with DNA repair defects. Although it is an important and interesting topic to detect reliable somatic SVs in a single cell, this study did not fully validate the PhaseDel algorithm. Thus, this reviewer thinks that the current validation data of the PhaseDel algorithm is not enough to convince researchers in the field of somatic mosaicism to use this new algorithm reliably due to the following concerns;

1. The authors implemented deep target-amplicon sequencing (TAseq) to validate the inferred deletions. However, only two cases were truly validated in multiple-tissue comparison (① ~50% VAF in affected scWGS, ② 0% in unaffected scWGS, ③ low-level (0.1-0.5%) VAF in bulk brain, ④ 0% in bulk peripheral tissue), as in Fig 2. In remaining 56 (out of 58) cases, it seems like they were validated only in MDA-amplified-single cell genomes. Overall, it is less convincing to claim that PhaseDel is accurate to discover true somatic deletions within phase-able reads of scWGS. The algorithm's effectiveness can be properly confirmed only when it is validated with external datasets or additional experiments. This reviewer suggests that they perform cross-validation by doing scWGS on MDA-amplified single fibroblasts and find somatic deletions in scWGS with PhaseDel, then verify the somatic deletions with bulk WGS of clones derived from the same single fibroblasts (① scWGS of MDA-amplified single fibroblast as PhaseDel query set, ② the same single-fibroblast-derived clones without MDA amplification as gold standards of true SVs). Or they may use public single-cell data from SRA database as suggested in SCCaller paper by Dong et al (Nat Methods, 2017).
2. The authors described that the somatic burdens of deletion increased with aging, but this observation was inconsistent with the previous Chronister WD's study (Cell Rep, 2019). The authors postulated that the discrepancy resulted from different size of deletion, which caused selective pressure for cell survival. In order to address this point adequately, the authors should show the analysis data about the aging trend of deletions of only larger size (megabase-scale). This reviewer also thinks that the locations (or sites) of deletion are important as the sizes of deletion. Can the authors provide the genomic locations

(exonic, intronic, genic, or intergenic) of nano-deletions and larger megabase-scale deletions, respectively? Is there any difference between them?

Minor comments

1. The authors classified the deletions into six categories according to DSB repair mechanisms (NHEJ, MMEJ, VNTR, NAHR, FoSTeS/MMBIR, TEI). Is there any deletion that can be classified as two or more classes simultaneously? Providing a Venn diagram of somatic deletion categories would be better to present the mutual exclusiveness of this classification.

2. Moreover, after constructing a class frequency vector for each cell and forming a matrix by combining them column-wisely, this reviewer suggests the authors to extract somatic deletion signatures using nonnegative matrix factorization (NMF)-based methods that are widely implemented in profiling somatic SNVs and short Indels.

3. The authors claimed that only about 25% of the genomic regions are phaseable, so an extrapolation process is required to proceed with statistical analysis on the somatic deletion burden. They constructed a model $T(c)=S(c)+E(c)=k_1+k_2 \exp(-k_3(c-2))$, where

$T(\cdot)$:the rate of all phased candidate, $S(\cdot)$:True somatic deletion rate, $E(\cdot)$:Erroneous FP rate

with respect to supporting read counts (c), assuming $E(\cdot)$ is exponentially decaying when c increases, then all parameters (k_1, k_2, k_3) were estimated with NLS curve fit. However, I doubt that the above equation is really the ideal model that best describes the true and false positives rates in this context. Have you ever compared multiple models with different equations and assumptions? Since many genomic variation callers utilize model-based probabilistic variant calling, likelihood modeling and maximum likelihood estimation of each parameter can be a good option (or its Bayesian extension).

4. It will be easier to comprehend the whole pipeline if the detailed process of phasing with gHet SNPs and filtering raw sets of GATK-called and DELLY2-called somatic deletions is summarized and presented as supplementary schematic flow.

5. This reviewer thinks the PhaseDel can be useful to find germline deletions too. It would be better to discuss this point at Discussion.

Reviewer #2 (Remarks to the Author):

In this study Kim et al. report a computational method that applies phasing with nearby germline variants to identify somatic deletions in individual neurons, which are otherwise difficult to detect. They identify an age-dependent increase in deletions that appears enriched in expressed genes, which parallels findings regarding single nucleotide mutations. They also profile neurons from individuals with mutations in DNA repair pathway genes and can detect both increases in numbers of small deletions and signatures of new mechanisms of interest. In particular, the discovery of deletion enrichment in expressed genes (and with transcription) and detection of increases in lesions that appear to be repaired by NHEJ are of interest. .

The paper is clear and authors provide convincing evidence for their core claims. The study design properly accounts for false positives/negatives, applying deep sequence validation along with reasonably stringent bioinformatic criteria. Unfortunately, because the LiRA method, as applied to deletions, can only account for roughly 25% of the genome, conclusions about certain mechanisms (such as the proposed role of double strand breaks in the promoters of neural early response genes) formally leave open the possibility that the 25% of the genome they survey does not proportionally represent the entire genome, though this seems unlikely to have major significance for the results as presented and is addressed by the authors. Overall this is a conceptually simple but technically strong study with novel and interesting conclusions that should be of interest to a broad audience of genome biologists and neuroscientists.

There are several minor considerations that should be addressed prior to publication.

1. Figure 1A: Red lines that are supposed to represent deletion artifacts are not visible in small version of the figure. Are the diagonal line-filled regions supposed to represent mapping of clipped reads like in 2A?
2. Line 110: Reference for microdeletions is needed.
3. Lines 114-116 – allowing one unphased read to decrease the false negative rate seems OK but it would be of interest to know the results without this addition for comparison.
4. Is the value of 205.3 (line 124) and 20.5 (line 125) the average calculated from several estimates from individual neurons? If so, it would be useful to also report the standard deviation. Additionally, a scatter plot of the parameter k_1 estimated in neurons from different individuals or all neurons in one individual could be shown to support the claim in the discussion that the estimated rate of deletions per cell varied much more than sSNV rates across individuals (lines 276-278).

5. Lines 140-141 – the authors found 2 deletions in bulk DNA samples – how many were tested? Would this provide an estimate of the number that are post-mitotic in neurons vs. arising in development?
6. Could the authors state what the data and fitting in Figure S1B represents? Is it for an individual cell?
7. In Figure 3A the authors observed somatic transposable element insertions (TEI category). it is not clear why TEIs are classified as deletions, or whether they were included in the PCR validation assay, which would not necessarily be sufficient to validate very large TEIs. This part was a bit confusing and could be clarified.
8. Figure 3 legend: in C state what the individual points represent (average from all neurons from one individual or each point is one neuron?). Some text related to panel D and the (D) label is missing.
9. The conclusions between line 244 – 248, that correspond to figures 4D and S3B are confusing and seem to contradict the in data 3E where there is a positive correlation between gene expression and NHEJ burden in normal neurons. Why is the NHEJ burden so high in Q1 of normal controls? Other than this data point at Q1, the curves for NHEJ in normal and CS NSCs seem similar. The colors of the dashed lines in the small version of 4D are difficult to distinguish.

Reviewer #3 (Remarks to the Author):

Review of Kim, et al.

“Prevalence and mechanisms of somatic deletions in single human neurons during normal aging and in DNA repair disorders”

The authors report a new bioinformatic tool, PhaseDel, which aims to identify genomic structural variants in single human neurons. The premise follows from this groups previous work, LiRA, wherein proximal heterozygous SNPs are employed to extend the veracity of mosaic variant calling in single neuronal genome datasets. The advance reported here is the identification 2,208 somatic “nanodeletions.” (3 bp – 113 Kbp, mean 193 bp, median 5 bp) among 108 neurons from 17 individuals. Of these, only 66 were selected for confirmation by direct sequencing. The authors go on to characterize DNA repair mechanisms associated with each nanodeletion, and conclude that neuronal microdeletions, relative to germline nanodeletions, are more frequently brought about by NHEJ DNA repair. While this is an important result, it is somewhat unsurprising.

I question whether this primary result, NHEJ predominance in neurons, reflects a form of observational bias. Are NHEJ-related nanodeletions simply more resilient to chimeric MDA artifacts and then more readily detected by PhaseDel?

I recommend that the manuscript is returned for substantial revision and additional experiments. The central question that remains to be addressed is if nanodeletion mechanisms, as predicted by PhaseDel, can be confirmed at similar frequencies for each DNA repair mechanism. Other critical elements are also enumerated below.

Primary critiques

1) The authors report 2208 nanodeletions, but examined only 66 for confirmation. A low bar for confidence is that at least 10% of candidates should be randomly selected for confirmation to assess the real value of PhaseDel. Moreover, this larger pool of candidates should be drawn from all 17 individuals in the dataset, examine candidates predicted to occur via different DNA repair mechanisms, and report confirmation rates for each mechanism and individual.

2) Confirmation of additional candidates should separately examine disease individuals. These results should also be reported separately for each individual and DNA repair mechanism, with all individuals tested.

3) Of the 66 candidates examined, 8 failed confirmation. The nature of these false positives should be reported. Again, to assess the value of PhaseDel it is critical to know whether failures were due to erroneous linked SNP calls or to questionable results from nucleotide sequencing of the predicted breakpoint?

4) The premise underlying PhaseDel's value is that single cell genome amplification via MDA can lead to chimeric amplicons and false positive genomic structural variants. However, the reported artifacts produced by MDA are primarily inversions, not small deletions. Following from point 3 above, reporting the nucleotide sequence at candidates that are not confirmed is critical to define the artifacts produced by Single Neuron MDA and to significantly advance the field.

Secondary critiques

1) Figures 1B and 1D, plot counts on Y axis rather than density. To evaluate PhaseDel, it is important to assess separately how many candidates are indicated by either GATK or DELLY.

2) Extrapolation of extant data to frequencies per neuron in Fig S1 and Results lines 119 – 125 is misleading. This is a speculative approach more appropriate for discussion.

Response to referee comments on “Prevalence and mechanisms of somatic deletions in single human neurons during normal aging and in DNA repair disorders”

We thank the reviewers for their comments, which we found highly constructive toward maximizing the clarity and accuracy of the manuscript. All the reviewers generally agreed on the importance of our findings and usefulness of our method, but raised two major concerns: (a) a lack of performance evaluation using independent data and (b) small validation size of predicted deletion candidates. In this revision, we further assessed the performance of PhaseDel using published kindred-clone data sets. We also significantly increased the number of deletion candidates for experimental validation using ultra-deep amplicon sequencing from 66 to 244 candidates, covering all individuals and disease status following the reviewer’s suggestion. Including these two major concerns, we have fully addressed all the comments below, and believe that their insights have significantly improved the quality and rigor our work and proposed method. In the main text, we have highlighted sections with major changes, including those describing the additional analyses and experimental validation.

Reviewer #1 (Remarks to the Author):

In this study, the authors developed the PhaseDel algorithm, which utilized GATK and DELLY2 to call somatic deletion candidates, then filtered and read-back phased, using clipped reads spanning reliable germline heterozygous SNP sites called from bulk WGS data. They then tried to identify true somatic deletions from 108 single neurons, which increased with age and in highly expressed genes in the human brain. Also, they showed that somatic deletions increased with distinctive patterns in neurons of individuals with DNA repair defects. Although it is an important and interesting topic to detect reliable somatic SVs in a single cell, this study did not fully validate the PhaseDel algorithm. Thus, this reviewer thinks that the current validation data of the PhaseDel algorithm is not enough to convince researchers in the field of somatic mosaicism to use this new algorithm reliably due to the following concerns;

1. The authors implemented deep target-amplicon sequencing (Taseq) to validate the inferred deletions. However, only two cases were truly validated in multiple-tissue comparison (① ~50% VAF in affected scWGS, ② 0% in unaffected scWGS, ③ low-level (0.1-0.5%) VAF in bulk brain, ④ 0% in bulk peripheral tissue), as in Fig 2. In remaining 56 (out of 58) cases, it seems like they were validated only in MDA-amplified-single cell genomes. Overall, it is less convincing to claim that PhaseDel is accurate to discover true somatic deletions within phase-able reads of scWGS. The algorithm’s effectiveness can be properly confirmed only when it is validated with external datasets or additional experiments. This reviewer suggests that they perform cross-validation by doing scWGS on MDA-amplified single fibroblasts and find somatic deletions in scWGS with PhaseDel, then verify the somatic deletions with bulk WGS of clones derived from the same single fibroblasts (① scWGS of MDA-amplified single fibroblast as PhaseDel query set, ② the same single-fibroblast-derived clones without MDA amplification as gold standards of true SVs). Or they may use public single-cell data from SRA database as suggested in SCCaller paper by Dong et al (Nat Methods, 2017).

→ We agree with the reviewer that validation using an external dataset is required to assess the accuracy of PhaseDel to detect true somatic deletions. As the reviewer suggested, we performed additional analysis using genome sequencing data of two MDA-amplified single fibroblasts (IL-11, IL-12) and their kindred clone (IL-1c) from Dong et al.¹ (Fig. S3A). Since phasing analysis can only cover

~25% of the genome and MDA-amplified single cell DNA has substantial allelic/locus dropouts, we cannot directly compare the deletion call sets generated from single cells and the unamplified clone to assess PhaseDel's performance. We rather focused on checking two important results, whether 1) the final PhaseDel deletion calls made from single cells are true somatic deletions that can also be observed in the kindred clone, and 2) the deletion rate estimated from single cells is consistent with the actual number of somatic deletions observed in the clone data.

For the first part, we checked deletion-supporting reads in the WGS of the unamplified clone for the PhaseDel deletion candidates that were obtained from the single fibroblast data (Fig. S3A). We considered a read to be deletion-supporting only if the read has both the same clipping/indel position and the clipped/indel sequences with the ones from the original single cell data, which ensures that they are representing the exact same event. The deletions with ≥ 5 read-depth at the breakpoint and ≥ 3 deletion-supporting reads from the unamplified clone data are considered as validated, resulting in 93.48% (43/46) and 94.74% (18/19) validation rates for IL-11 and IL-12 PhaseDel deletion sets, respectively (Fig. S3B).

For the second part, from the clone data we counted the number of somatic deletions that are not observed in any other data except for the kindred groups, to obtain kindred-cell-specific deletions (Fig. S3A, red stars). To reduce artifactual errors from the call set, we excluded all the candidates that 1) matched with known polymorphic deletions reported in the 1000 Genomes Project or the Database of Genomic Variants (DGV), 2) have more than one deletion-supporting read from the matched bulk WGS, or 3) have more than one deletion-supporting read in bulk WGS of the unrelated individuals from our study or any single-cell WGS data from the non-kindred group of Dong et al. and our study (Fig. S3C). This filtration may also remove some true clonal deletions shared with non-kindred cells; however, based on our validation experiments, most phased deletions we found were likely non-clonal (207/209, including new validation results in this revision) and therefore we considered the loss of clonal deletion would be negligible for the overall rate estimation. We found 452 somatic deletion candidates from the clone data after all filtering steps, which is consistent with the rates estimated from the kindred single cells (Fig. S3D; 468.66 and 407.25 for IL-11 and IL-12, respectively). We revised the main text and added supplementary figure 3 to describe the details of the validation analysis using the kindred system.

2. The authors described that the somatic burdens of deletion increased with aging, but this observation was inconsistent with the previous Chronister WD's study (Cell Rep, 2019). The authors postulated that the discrepancy resulted from different size of deletion, which caused selective pressure for cell survival. In order to address this point adequately, the authors should show the analysis data about the aging trend of deletions of only larger size (megabase-scale). This reviewer also thinks that the locations (or sites) of deletion are important as the sizes of deletion. Can the authors provide the genomic locations (exonic, intronic, genic, or intergenic) of nano-deletions and larger megabase-scale deletions, respectively? Is there any difference between them?

→ The reviewer requested to see the comparison between nano-deletions and megabase-scale deletions for their trends with age and their distribution of genomic locations. Unfortunately, our algorithm targets smaller deletions with clear single-base-level breakpoints (maximum size 100 Kbp, mean size 155 bp in this study), thus it cannot cover megabase-scale deletions. Such large deletions are typically detected by read-depth-based analysis of large genomic bins (>10 Kbp), and multiple displacement amplification (MDA) used for single-cell whole-genome amplification in this study is

generally considered not suitable for large CNV detection due to its low uniformity of amplification across the genome². The extremely low occurrence of megabase-scale CNVs in a single neuron (0.32 CNVs per neuron from Chronister et al.³) and limited genomic regions applicable to phasing analysis (~25% of the genome with nearby germline heterozygous SNPs) also raise practical difficulties in obtaining the megabase-scale deletion set; even if our method was able to detect megabase-scale deletions, the expected number would be around ~8.64 deletions ($0.32 \times 0.25 \times 108$ normal neurons), which is too small to perform any statistical tests to analyze trends with age or with their genomic distribution.

The major reason the reviewer suggested such comparisons is to explain the inconsistency between our observation and the previous reports in Chronister et al. However, we note that both studies actually demonstrate common major findings: 1) higher burden of CNVs (deletions) in the aged group and 2) enrichment of CNVs in neuronal genes. Chronister et al. reported anti-correlation between age and the fraction of neurons harboring CNVs, not the number of CNVs per neuron. For neurons harboring CNVs, they also observed a higher number of CNVs per cell in the aged group as we observed, although their observation was not statistically significant due to the low and highly variable numbers of CNVs. For example, the 24-year-old and the 86-year-old groups in their data had 38.9% and 4% of neurons with CNVs, but the average number of CNVs per CNV-harboring-neuron were 2.8 and 5.0 respectively.

Chronister et al. also reported significant enrichment of CNVs in long genes with >100Kbp length that are neuronally expressed with roles in neuronal connectivity and synaptic plasticity^{4, 5, 6}. They also emphasized the identification of CNV hotspots in well-known neuronal genes commonly reported in all three previous studies^{4, 5, 6} and suggested that gene transcription is the main cause of neuronal CNVs, as we found in our analysis. Both Chronister et al. and our study showed the accumulation of deletions (CNVs) with age in a neuron and their enrichment in neuronally expressed genes, and the only difference is the decreasing fraction of CNV-harboring neurons with age for megabase-scale CNVs observed in Chronister et al. We expect that accumulating megabase-scale CNVs will exert significantly larger selective pressure for cell survival than small deletions, resulting in selective loss of CNV-harboring neurons with age but not for small deletions. We added a more detailed description of the comparison between Chronister et al. and our findings in the Discussion section.

Minor comments

1. The authors classified the deletions into six categories according to DSB repair mechanisms (NHEJ, MMEJ, VNTR, NAHR, FoSTeS/MMBIR, TEI). Is there any deletion that can be classified as two or more classes simultaneously? Providing a Venn diagram of somatic deletion categories would be better to present the mutual exclusiveness of this classification.

→ All deletion types are exclusive based on their selection criteria, so there is no deletion classified as more than one class. The only exceptions are deletions with 1-3 bp homology between their breakpoints, which might originate from either NHEJ or MMEJ. These deletions are classified as the unknown group (MH=1,2,3) and excluded from further analyses. We newly added a flowchart to describe the classification process and the relationship between different groups in detail (Fig. S2B).

2. Moreover, after constructing a class frequency vector for each cell and forming a matrix by

combining them column-wisely, this reviewer suggests the authors to extract somatic deletion signatures using nonnegative matrix factorization (NMF)-based methods that are widely implemented in profiling somatic SNVs and short Indels.

→ As the reviewer suggested, we performed NMF-based analysis for somatic deletion signatures (see figure below). We first constructed a contingency table (6 DSB repair mechanisms×157 total neurons) and tested different numbers of signatures. Cophenetic correlation coefficient indicated two signatures as the optimal number for extraction (the smallest rank for which cophenetic correlation coefficient starts decreasing in Fig. A below). The first signature was almost exclusive for NHEJ, and the second signature was composed mainly of MMEJ and NHEJ (Fig. B). Relative contribution of two signatures for each cell is presented in Fig. C.

However, signature analysis showed no further results from the current analyses already presented in the paper. We think there are two potential reasons for this: 1) few somatic deletions other than NHEJ or MMEJ and 2) low numbers of deletions per cell (16.4 high-confidence deletions per cell on average). Such uniform and sparse distribution of somatic deletions resulted in the extraction of very simple signatures just composed of two mechanisms, NHEJ and MMEJ, and the comparison of signature contributions showed no difference from the burden analysis for each mechanism that we had presented in Fig. 3 and 4. Therefore we have decided not to include this analysis in the current manuscript. Future improvements in somatic deletion detection and its application to multiple cell types and disease contexts will reveal a larger number of deletions across various underlying mechanisms and provide more insights into mutational processes.

3. The authors claimed that only about 25% of the genomic regions are phaseable, so an extrapolation process is required to proceed with statistical analysis on the somatic deletion burden. They constructed a model $T(c)=S(c)+E(c)=k_1+k_2 \exp(-k_3 (c-2))$, where $T(c)$:the rate of all phased candidate, $S(c)$:True somatic deletion rate, $E(c)$:Erroneous FP rate with respect to supporting read counts (c), assuming $E(c)$ is exponentially decaying when c increases, then all parameters (k_1,k_2,k_3) were estimated with NLS curve fit. However, I doubt that the above equation is really the ideal model that best describes the true and false positives rates in this context. Have you ever compared multiple models with different equations and assumptions? Since many genomic variation callers utilize model-based probabilistic variant calling, likelihood modeling and maximum likelihood estimation of each parameter can be a good option (or its Bayesian extension).

→ We appreciate the reviewer's suggestion and agree with the reviewer that more sophisticated modeling would improve the deletion rate estimation process. We adopted Bayesian inference using Markov chain Monte Carlo (MCMC) sampling to estimate the parameters of the suggested model. We implemented the mixture model and conducted MCMC simulation using the R rstan package, composed of four MCMC chains with 5,000 burn-in steps and 10,000 iterations. This process estimates not only the optimal parameter values (posterior means) but also their confidence intervals, providing additional information of estimation reliability. Newly estimated rates were fairly consistent with the previous estimates (Pearson $r=0.91$, $p<2.2\times 10^{-16}$; see figure below) and hold the same conclusions for all the tests in our study. The power-law distribution was also tested for the decay function of the error model, but it was discarded due to the large number of samples that failed to converge. We described the revised rate estimation process in detail in the Methods section.

4. It will be easier to comprehend the whole pipeline if the detailed process of phasing with gHet SNPs and filtering raw sets of GATK-called and DELLY2-called somatic deletions is summarized and presented as supplementary schematic flow.

→ We added a schematic flow diagram for the overall PhaseDel workflow in Supplementary Figure S2A.

5. This reviewer thinks the PhaseDel can be useful to find germline deletions too. It would be better to discuss this point at Discussion.

→ We thank the reviewer for the suggestion. We additionally discussed the application of PhaseDel for germline variant detection in the Discussion section as follows: “PhaseDel will also be useful for the detection of germline deletions, for example, those in complex regions such as human leukocyte antigen (HLA) loci as the linkage between deletion-supporting reads and other adjacent germline variants will help reduce false positive predictions.”

Reviewer #2 (Remarks to the Author):

In this study Kim et al. report a computational method that applies phasing with nearby germline variants to identify somatic deletions in individual neurons, which are otherwise difficult to detect. They identify an age-dependent increase in deletions that appears enriched in expressed genes, which parallels findings regarding single nucleotide mutations. They also profile neurons from individuals with mutations in DNA repair pathway genes and can detect both increases in numbers of small deletions and signatures of new mechanisms of interest. In particular, the discovery of deletion enrichment in expressed genes (and with transcription) and detection of increases in lesions that appear to be repaired by NHEJ are of interest.

The paper is clear and authors provide convincing evidence for their core claims. The study design properly accounts for false positives/negatives, applying deep sequence validation along with reasonably stringent bioinformatic criteria. Unfortunately, because the LiRA method, as applied to deletions, can only account for roughly 25% of the genome, conclusions about certain mechanisms (such as the proposed role of double strand breaks in the promoters of neural early response genes) formally leave open the possibility that the 25% of the genome they survey does not proportionally represent the entire genome, though this seems unlikely to have major significance for the results as presented and is addressed by the authors. Overall this is a conceptually simple but technically strong study with novel and interesting conclusions that should be of interest to a broad audience of genome biologists and neuroscientists.

There are several minor considerations that should be addressed prior to publication.

1. Figure 1A: Red lines that are supposed to represent deletion artifacts are not visible in small version of the figure. Are the diagonal line-filled regions supposed to represent mapping of clipped reads like in 2A?

→ We apologize for the insufficient visibility of the figures. We increased the thickness of the red lines indicating deletion artifacts in Figure 1A. The reviewer was correct that the diagonal line-filled regions represent the clipped part of the read. We added a label and a description for this to the figure and to the figure legend, respectively, for clarity.

2. Line 110: Reference for microdeletions is needed.

→ We thank the reviewer for pointing this out. A reference for microdeletions, Carvill et al., Curr Opin

Genet Dev 2013, has been added.

3. Lines 114-116 – allowing one unphased read to decrease the false negative rate seems OK but it would be of interest to know the results without this addition for comparison.

→ We added Supplementary Figure S1E to show the detected fractions of germline deletions with and without allowing one unphased read. The comparison shows that we missed an additional 15% of true germline deletions when not allowing any unphased read, resulting in an average sensitivity of less than 70%.

4. Is the value of 205.3 (line 124) and 20.5 (line 125) the average calculated from several estimates from individual neurons? If so, it would be useful to also report the standard deviation. Additionally, a scatter plot of the parameter k_1 estimated in neurons from different individuals or all neurons in one individual could be shown to support the claim in the discussion that the estimated rate of deletions per cell varied much more than sSNV rates across individuals (lines 276-278).

→ The reviewer is correct that these values are the averages from individual neurons. We added standard deviations in the text. We also added a boxplot with jitter points (Fig. S6D) to show the distribution of sSNV and deletion rate per cell per individual, as the reviewer suggested. Since the estimated sSNV and deletion rates are in different scales, each value was normalized by the average rate of each individual to make a proper comparison. As we previously described, the estimated deletion rates generally exhibited greater dispersion than sSNV rates.

5. Lines 140-141 – the authors found 2 deletions in bulk DNA samples – how many were tested? Would this provide an estimate of the number that are post-mitotic in neurons vs. arising in development?

→ Including additional validation experiments added to this revision (see the response to Primary Critique 1 of Reviewer 3), only the same two deletions out of 209 validated deletions as previously reported were validated with unamplified bulk DNA. The estimation of the ratio between post-mitotic, i.e., single-cell-specific vs. developmental deletions is an intriguing question to answer; however, we think it is too challenging to estimate based on the presence or absence of the events in bulk DNA amplicon sequencing. The reason is that there might be many factors that affect detection power, for example, sequencing depths at validation sites and background noise level of amplicon sequencing data. Furthermore, the number of clonal deletions is too small to make a reliable estimation.

6. Could the authors state what the data and fitting in Figure S1B represents? Is it for an individual cell?

→ The reviewer is correct that the model in Figure S1B (now it is Fig. S1G) is for an individual cell. We apologize for the insufficient description. We provided more details in the 'estimation of somatic deletion rate' section in the main text as well as the legend of Fig. S1G. Briefly, from one given single cell sequencing data, we usually obtain more than hundreds of initial deletion candidates and need to determine which of them are true somatic deletions even from the phased ones and estimate how many deletions are actually present in a given cell. Figure S1B (now Fig. S1G) represents this estimation and filtration process from a single cell. All deletion candidates are first grouped into many subgroups

based on their linked-read counts between a deletion and a nearby germline heterozygous SNP. Deletion rates are calculated separately for each subgroup and displayed together (black dots) to check their distribution along with the supporting linked-read counts; each black dot represents a genome-wide rate for a subset of deletions that had the same linked-read counts. Their distribution is made of a mixture of true somatic deletions and false positive errors, and the errors are expected to decrease with the linked-read counts as more perfectly linked reads are less likely to be caused by errors. We fit a two-component model consisting of a constant for true deletions and a decaying function for errors, and use the parameters of components to estimate the final deletion rate (the constant, blue line) and FDR (the ratio between area under two fitted curves).

7. In Figure 3A the authors observed somatic transposable element insertions (TEI category). it is not clear why TEIs are classified as deletions, or whether they were included in the PCR validation assay, which would not necessarily be sufficient to validate very large TEIs. This part was a bit confusing and could be clarified.

→ The TEI category of *deletion* calls is indeed confusing because of the term ‘insertion’ in the name as adopted from Yang et al.⁷ This type of events is technically a deletion in a WGS-profiled sample relative to the reference genome; however, as the name indicates, TE mobilization/insertion is the mechanism underlying the presence of the TE sequence in reference samples whose DNA was analyzed to create the initial reference genome assembly. The criteria for TEI deletion classification we adopted from Yang et al.⁷ requires >80% of a deletion region and >80% of a known TE sequence to be overlapped to ensure TE mobilization in the reference sample as the most-likely mechanism underlying the event.

This category is a dominant mechanism of germline deletions (~50% of deletions), which represents many polymorphic TE loci that are present in the reference genome but absent in a given sample as shown in the top left panel of Fig. 3A and Fig. 2 in Yang et al. However, this type of events is unlikely to happen in a somatic state unless rare genomic rearrangements, such as Alu-Alu or LTR-LTR recombination (Alu- or LTR-mediated deletion)^{8,9} delete a reference TE copy (Fig. 3A, bottom panel). We had a total of two somatic deletions annotated as TEI, but our validation amplicon sequencing failed to get any reads from those regions even for the wildtype allele without the deletion, so the nature of the two events is inconclusive.

8. Figure 3 legend: in C state what the individual points represent (average from all neurons from one individual or each point is one neuron?). Some text related to panel D and the (D) label is missing.

→ We added a description that each point represents a single neuron in the legend of Figure 3 and fixed the missing label of panel D.

9. The conclusions between line 244 – 248, that correspond to figures 4D and S3B are confusing and seem to contradict the in data 3E where there is a positive correlation between gene expression and NHEJ burden in normal neurons. Why is the NHEJ burden so high in Q1 of normal controls? Other than this data point at Q1, the curves for NHEJ in normal and CS NSCs seem similar. The colors of the dashed lines in the small version of 4D are difficult to distinguish.

→ We revised the main text to improve the clarity of the multiple comparisons we have performed in

Figure 4D. The pattern of the dashed lines was also updated for a better distinction. All deletion burden analyses with gene expression in our study showed consistent patterns of the increased NHEJ burden along with gene expression levels. But this positive correlation was observed only if we used proper pairs of single-cell WGS and gene expression data, i.e., those generated in a similar context. For example, the NHEJ deletion burden from PFC neurons of CS patients increased with gene expression levels from iPSC-derived NSCs of CS patients (Fig. 4D, solid lines). However, the positive relationship was not observed as expected when comparing the same deletion burden of CS PFC neurons with gene expression levels from iPSC-derived NSCs of *normal* individuals (Fig. 4D, dashed lines). These results further support that the NHEJ deletions largely occur by gene transcription.

The reviewer seemed to have misread that the dashed line in Fig. 4D represents a comparison of the deletion burden and gene expression in normal neurons and thought that it contradicted the results in Fig. 3E. But the two figures show different comparisons: deletions from normal PFC neurons with gene expression level from normal PFC (Fig. 3E) and deletions from CS PFC neurons with gene expression levels in normal iPSC-NSC (Fig. 4D, dashed lines). The high NHEJ burden from CS PFC neurons in Q1 genes from normal iPSC-NSC suggests that the gene expression patterns between iPSC-NSC from CS patients and control individuals are quite different. We hope that the revised text clearly describes the comparisons.

Reviewer #3 (Remarks to the Author):

The authors report a new bioinformatic tool, PhaseDel, which aims to identify genomic structural variants in single human neurons. The premise follows from this groups previous work, LiRA, wherein proximal heterozygous SNPs are employed to extend the veracity of mosaic variant calling in single neuronal genome datasets. The advance reported here is the identification 2,208 somatic “nanodeletions.” (3 bp – 113 Kbp, mean 193 bp, median 5 bp) among 108 neurons from 17 individuals. Of these, only 66 were selected for confirmation by direct sequencing. The authors go on to characterize DNA repair mechanisms associated with each nanodeletion, and conclude that neuronal microdeletions, relative to germline nanodeletions, are more frequently brought about by NHEJ DNA repair. While this is an important result, it is somewhat unsurprising.

I question whether this primary result, NHEJ predominance in neurons, reflects a form of observational bias. Are NHEJ-related nanodeletions simply more resilient to chimeric MDA artifacts and then more readily detected by PhaseDel?

I recommend that the manuscript is returned for substantial revision and additional experiments. The central question that remains to be addressed is if nanodeletion mechanisms, as predicted by PhaseDel, can be confirmed at similar frequencies for each DNA repair mechanism. Other critical elements are also enumerated below.

→ We thank the reviewer for raising an important point about potential observational bias of nanodeletions across different mechanisms. In addition to validating a significantly more deletion candidates (see the response to Primary critique 1), we examined if the filtering rates of PhaseDel phasing analysis vary across different mechanisms. Briefly, PhaseDel deletion calling includes three major steps: i) collecting initial candidates using GATK and DELLY2, including germline, somatic, and false positive events, ii) linkage analysis to remove false positives, including chimeric MDA artifacts,

and iii) selecting somatic candidates and annotating deletion mechanisms according to previously reported criteria⁷. Among the steps, the linkage analysis is where PhaseDel might cause the observational bias. We found overall the proportions of deletion candidates across mechanisms were comparable before and after the phasing-based linkage analysis having NHEJ deletions consistently as the dominant mechanism (Fig. S4C). Importantly, a similar fraction of candidates remained for each mechanism except for MMEJ and NAHR (Fig. S4D), supporting against the observational bias that favors NHEJ. Both MMEJ and NAHR have sequence homology between the two deletion breakpoints, which is also an important feature for the formation of chimeric artifacts. Therefore, linkage analysis is likely to filter out a higher fraction of candidates as false positives for these two mechanisms.

Our observation of no somatic deletions by FoStES and NAHR is expected as both mechanisms are known to occur during DNA replication¹⁰, a process absent in post-mitotic neurons. It is also natural to have few somatic deletion candidates of the TEI category as genomic rearrangements involving reference TE copies occur very rarely in somatic cells. Furthermore, for germline deletions, we obtained similar mechanistic contributions to the previous work⁷ (Fig. 3A) supporting that our mechanism prediction works as expected. We described these additional analyses that support against the observation bias of PhaseDel in the section “Somatic nanodeletions increase with age and reflect distinctive underlying repair mechanisms”.

Primary critiques

1) The authors report 2208 nanodeletions, but examined only 66 for confirmation. A low bar for confidence is that at least 10% of candidates should be randomly selected for confirmation to assess the real value of PhaseDel. Moreover, this larger pool of candidates should be drawn from all 17 individuals in the dataset, examine candidates predicted to occur via different DNA repair mechanisms, and report confirmation rates for each mechanism and individual.

→ Following the reviewer’s suggestion, we attempted to perform amplicon sequencing to validate additional 197 candidates across all 17 individuals and mechanisms, and achieved an overall validation rate of 85.7%. After improving our rate estimation module (please refer to the minor point (Q3) of reviewer 1), we obtained a total of 1,751 high-confidence deletion candidates. Among them, we selected additional 197 candidates across 17 normal individuals and different deletion mechanisms for validation. After generating the amplicon sequencing data, we found that 19 of them failed to generate any reads even for the wild-type allele without a deletion, and excluded them from the validation set. Unfortunately, this resulted in no candidates for two normal individuals (5559 and 5943). The two individuals did not show any notable deletion counts different from the other individuals (Table S2). Out of 244 candidates including the previously validated ones, 209/244 (85.7%) were validated with MDA-amplified single-cell DNA, resulting into a similar rate to the previous result (87.9%). The validation rates across individuals and mechanisms were overall very high with an average validation rate of 96%. Only three individuals (5532, 5840, 5657) showed 62.5%, 60%, and 60%, respectively (Fig. S4A). We think that our validation rate estimated from the experimental validation of >10% candidates demonstrates the accuracy of PhaseDel.

In terms of the mechanism coverage, note that only NHEJ and MMEJ deletions were available for validation with the following reasons. We initially had two FoStES candidates, which require the presence of >10bp insertion at the deletion breakpoint. However, visual inspection of these candidates found that both had insertions in the other alleles from the deletions suggest inaccurate annotation of FoStES. We further updated our annotation module to check the linkage between the insertion and

deletion to determine FoStES candidates, and no longer annotate them as FoStES. We also had two TEI deletion candidates, but both of them failed to generate any reads with amplicon sequencing. Lastly, we do not have any somatic candidates for NAHR and VNTR.

2) Confirmation of additional candidates should separately examine disease individuals. These results should also be reported separately for each individual and DNA repair mechanism, with all individuals tested.

→ We tested a total of 92 additional candidates for all 11 diseased individuals (Fig. S4B). Most diseased individuals had high validation rates for both NHEJ and MMEJ mechanisms, with the exception of two XP individuals (5379 and 5316). Since this was also the case for some normal individuals such as 5657 and 5840, we think that validation rates vary among individuals rather than disease status.

3) Of the 66 candidates examined, 8 failed confirmation. The nature of these false positives should be reported. Again, to assess the value of PhaseDel it is critical to know whether failures were due to erroneous linked SNP calls or to questionable results from nucleotide sequencing of the predicted breakpoint?

→ We thank the reviewer for the suggestion. Out of 244 validation candidates, 35 candidates including the previous 8 candidates, failed to be validated in MDA-amplified DNA. 5 out of them had deletion-supporting reads in the negative control, so considered not validated. For the remaining 30 failed candidates, we examined multiple features including deletion types, supporting read count, genomic context, and homology sequence length to identify representative features that differed from the validated candidates, but none of the examined features showed differences. We expect that they are likely due to the discrepant sampling of deletion supporting reads between the scWGS and amplicon sequencing data, especially for the MDA artifacts that occurred late during MDA amplification and thus the amplified library has only a limited number of DNA templates carrying the artifacts. We described our attempt to characterize the validation failure in the manuscript.

4) The premise underlying PhaseDel's value is that single cell genome amplification via MDA can lead to chimeric amplicons and false positive genomic structural variants. However, the reported artifacts produced by MDA are primarily inversions, not small deletions. Following from point 3 above, reporting the nucleotide sequence at candidates that are not confirmed is critical to define the artifacts produced by Single Neuron MDA and to significantly advance the field.

→ The reviewer is correct that inversion is the most common form of MDA artifacts as previous studies including our own work have reported^{11,12}. However, previous studies did not consider small deletions as artifactual consequences of the MDA process. For example, Lasken et al.¹² identified chimeric artifacts by only selecting partially mapped reads that were divided into two segments, and this selection did not include the reads with small deletions. They reported that more than 85% of identified chimeras were inverted, but also found that they were with intervening deletions, showing the prevalence of deletions as MDA artifacts. In our previous work¹¹, we considered deletions only based on discordant reads, which had the minimum size of ~300 bp between 3' ends and did not include the reads with small deletions. Therefore, the prevalence of small deletions generated during the MDA process has not yet been analyzed. Polymerase slippage is the most common mechanism for

short indel generation (that includes small deletions)^{13, 14} during DNA synthesis regardless of *in vivo* DNA replication or *in vitro* genome amplification, and it might result in the large number of small deletion candidates we have observed. Since these polymerase-slippage-derived artifacts can be filtered out by the linkage analysis of PhaseDel, we think that the value of our method in accurate variant detection in scWGS data is significant.

Secondary critiques

1) Figures 1B and 1D, plot counts on Y axis rather than density. To evaluate PhaseDel, it is important to assess separately how many candidates are indicated by either GATK or DELLY.

→ We are grateful for the reviewer for pointing this out. We first have plotted the Figure 1B and 1D with actual counts in Supplementary Figure S1A and S1D. As shown in the figures, there are far more events in one group than the other, i.e., more GATK calls than DELLY calls and more germline events than somatic ones, and thus, plotting the counts from both groups on the same scale makes it difficult to see the patterns of the under-represented groups (Suppl. S1A and S1D). Since we intend to demonstrate that GATK and DELLY overall target deletions of different size ranges and thus utilizing both of them is beneficial to capture a broader spectrum of deletions in Fig. 1B, and that the size distribution is different between germline and somatic deletions in Fig. 1D, we present the density plots as main figures and count plots as supplementary figures. In addition to Fig. S1A and S1D, we thought that the reviewer wanted to check actual numbers of candidates generated from GATK and DELLY for both initial call sets and the phased sets, we added Supplementary Figure S1B and S1C. GATK had far more deletion candidates than DELLY in the initial call sets, but the linkage analysis filtered out a larger portion of GATK calls thus resulted in the increased proportion of DELLY calls in the phased sets.

2) Extrapolation of extant data to frequencies per neuron in Fig S1 and Results lines 119 – 125 is misleading. This is a speculative approach more appropriate for discussion.

→ To the best of our knowledge, none of single-cell or single-molecule whole-genome sequencing can cover the entire genome due to substantial allelic/locus dropouts or limited tagging sites, such as Tn5 transposase and restriction enzyme target sites. Therefore, all previous studies perform model-based estimation based on reliable partial genomic regions of sequencing data to obtain genome-wide rates^{15, 16, 17}. To support the reliability of our estimation process, we performed additional analysis using an external dataset of two single fibroblasts and their kindred clones in this revision (please refer to the response to the first comment of reviewer 1 for details). We obtained comparable somatic deletion rates estimated from two different single cells (468.66 and 407.25) to the actual number of somatic deletions without any rate extrapolation in an unamplified bulk clone (452), supporting the reliability of our estimation process. We described this additional validation of our approach in the 'Evaluation of PhaseDel accuracy and validation using ultra-deep amplicon sequencing' section with the Supplementary Figure S3.

References

1. Dong X, *et al.* Accurate identification of single-nucleotide variants in whole-genome-amplified single cells. *Nat Methods* **14**, 491-493 (2017).
2. Mallory XF, Edrisi M, Navin N, Nakhleh L. Methods for copy number aberration detection from single-cell DNA-sequencing data. *Genome Biol* **21**, 208 (2020).
3. Chronister WD, *et al.* Neurons with Complex Karyotypes Are Rare in Aged Human Neocortex. *Cell Rep* **26**, 825-835 e827 (2019).
4. King IF, *et al.* Topoisomerases facilitate transcription of long genes linked to autism. *Nature* **501**, 58-62 (2013).
5. Wei PC, *et al.* Long Neural Genes Harbor Recurrent DNA Break Clusters in Neural Stem/Progenitor Cells. *Cell* **164**, 644-655 (2016).
6. Zylka MJ, Simon JM, Philpot BD. Gene length matters in neurons. *Neuron* **86**, 353-355 (2015).
7. Yang L, *et al.* Diverse mechanisms of somatic structural variations in human cancer genomes. *Cell* **153**, 919-929 (2013).
8. Kim S, Cho CS, Han K, Lee J. Structural Variation of Alu Element and Human Disease. *Genomics Inform* **14**, 70-77 (2016).
9. Thomas J, Perron H, Feschotte C. Variation in proviral content among human genomes mediated by LTR recombination. *Mob DNA* **9**, 36 (2018).
10. Hastings PJ, Ira G, Lupski JR. A microhomology-mediated break-induced replication model for the origin of human copy number variation. *PLoS Genet* **5**, e1000327 (2009).
11. Evrony GD, *et al.* Cell lineage analysis in human brain using endogenous retroelements. *Neuron* **85**, 49-59 (2015).
12. Lasken RS, Stockwell TB. Mechanism of chimera formation during the Multiple Displacement Amplification reaction. *BMC Biotechnol* **7**, 19 (2007).
13. Montgomery SB, *et al.* The origin, evolution, and functional impact of short insertion-deletion variants identified in 179 human genomes. *Genome Res* **23**, 749-761 (2013).
14. Taylor MS, Ponting CP, Copley RR. Occurrence and consequences of coding sequence insertions and deletions in Mammalian genomes. *Genome Res* **14**, 555-566 (2004).
15. Bohrsen CL, *et al.* Linked-read analysis identifies mutations in single-cell DNA-sequencing data. *Nat Genet* **51**, 749-754 (2019).
16. Luquette LJ, Bohrsen CL, Sherman MA, Park PJ. Identification of somatic mutations in single cell DNA-seq using a spatial model of allelic imbalance. *Nat Commun* **10**, 3908 (2019).
17. Xing D, Tan L, Chang CH, Li H, Xie XS. Accurate SNV detection in single cells by transposon-based whole-genome amplification of complementary strands. *Proc Natl Acad Sci U S A* **118**, (2021).

REVIEWERS' COMMENTS

Reviewer #1 (Remarks to the Author):

The authors addressed this reviewer's concerns well by performing additional analyses. This reviewer agrees to that this study provides a new insight of somatic deletion in neurons during aging.

Reviewer #2 (Remarks to the Author):

The authors have now satisfactorily addressed my key concerns and I support publication of the manuscript.

Reviewer #3 (Remarks to the Author):

My concerns were addressed. This is very interesting work, I recommend accepting the manuscript with no further revisions.

All the reviewers agreed to publish our manuscript with no further revisions. Again, we thank all the reviewers for their careful reading and previous valuable suggestions.

REVIEWERS' COMMENTS

Reviewer #1 (Remarks to the Author):

The authors addressed this reviewer's concerns well by performing additional analyses. This reviewer agrees to that this study provides a new insight of somatic deletion in neurons during aging.

Reviewer #2 (Remarks to the Author):

The authors have now satisfactorily addressed my key concerns and I support publication of the manuscript.

Reviewer #3 (Remarks to the Author):

My concerns were addressed. This is very interesting work, I recommend accepting the manuscript with no further revisions.